# Aion is a bistable anion-conducting channelrhodopsin that provides temporally extended and reversible neuronal silencing

Silvia Rodriguez-Rozada[1], Jonas Wietek [2,3,4], Federico Tenedini[5], Kathrin Sauter[1,5], Neena Dhiman[6,7], Peter Hegemann [2], Peter Soba[5,6,7] & J. Simon Wiegert [1✉]

Optogenetic silencing allows to reveal the necessity of selected neuronal populations for various neurophysiological functions. These range from synaptic transmission and coordinated neuronal network activity to control of specific behaviors. An ideal single-component optogenetic silencing tool should be switchable between active and inactive states with precise timing while preserving its activity in the absence of light until switched to an inactive state. Although bistable anion-conducting channelrhodopsins (ACRs) were previously engineered to reach this goal, their conducting state lifetime was limited to only a few minutes and some ACRs were not fully switchable. Here we report Aion, a bistable ACR displaying a long-lasting open state with a spontaneous closing time constant close to 15 min. Moreover, Aion can be switched between the open and closed state with millisecond precision using blue and orange light, respectively. The long conducting state enables overnight silencing of neurons with minimal light exposure. We further generated trafficking-optimized versions of Aion, which show enhanced membrane localization and allow precisely timed, long-lasting all-optical control of nociceptive responses in larvae of *Drosophila melanogaster*. Thus, Aion is an optogenetic silencing tool for inhibition of neuronal activity over many hours which can be switched between an active and inactive state with millisecond precision.

[1] Research Group Synaptic Wiring and Information Processing, Center for Molecular Neurobiology Hamburg, University Medical Center Hamburg-Eppendorf, 20251 Hamburg, Germany. [2] Institute for Biology, Experimental Biophysics, Humboldt University Berlin, D-10115 Berlin, Germany. [3] Department of Brain Sciences, Weizmann Institute of Science, Rehovot 76100, Israel. [4] Department of Molecular Neuroscience, Weizmann Institute of Science, Rehovot 76100, Israel. [5] Research Group Neuronal Patterning and Connectivity, Center for Molecular Neurobiology Hamburg, University Medical Center Hamburg-Eppendorf, 20251 Hamburg, Germany. [6] LIMES Institute, Department of Molecular Brain Physiology and Behavior, University of Bonn, 53115 Bonn, Germany. [7] Institute of Physiology and Pathophysiology, Friedrich-Alexander-Universität Erlangen-Nürnberg, 91054 Erlangen, Germany. ✉email: simon.wiegert@zmnh.uni-hamburg.de

Optogenetic manipulation of neural activity is a powerful approach to assess the function of defined neuronal populations from the subcellular to the behavioral level. While optogenetic neuronal excitation is relatively straightforward, optogenetic silencing of neural activity is limited by several constraints[1,2]. For example, suppression of neuronal activity over extended time periods requires either continuous illumination or extended activity of the inhibitory tool in absence of light.

Suppression of neuronal activity by light-driven microbial ion pumps such as Archaerhodopsin (Arch)[3] and Halorhodopsin (NpHR)[4] is a well-established approach. These opsins hyperpolarize neurons in response to light by actively transporting protons out of (Arch) or chloride ions into (NpHR) the cell. However, rhodopsin pumps demand continuous high-intensity illumination for efficient neuronal silencing due to their one-ion per photon transport ratio. Moreover, as demonstrated for NpHR, they can inactivate when illuminated for prolonged time periods[5], which, together with their influence on local ion homeostasis[6,7], limits their applicability for long-term silencing of neurons.

The development of light-activated anion-conducting channelrhodopsins (ACRs) by targeted mutagenesis of cation-conducting channelrhodopsins (CCRs)[8–12] and the discovery of natural ACRs[13] introduced a new class of inhibitory optogenetic tools that overcame some of the limitations of ion pumps. Upon illumination, ACRs silence neuronal activity by shunting membrane voltage to the reversal potential of chloride ($E_{Cl^-}$), similar to natural ionotropic GABA$_A$ receptors[10]. However, fast-cycling channels still require continuous illumination to maintain a conducting state.

Therefore, optogenetic silencing for extended time periods on the scale of minutes to hours requires ACRs with slow off-kinetics of the conducting state and no accumulation of late non-conducting states. Additionally, they need to be switchable to their non-conducting dark-state to enable temporally precise on- and offsetting of the silencing period. ACRs with slow off-kinetics are furthermore relevant in long-term optogenetic experiments to prevent tissue-heating and phototoxicity by continuous illumination, particularly in the short wavelength range[14,15]. The slowed-down photocycle greatly enhances the operational light sensitivity of engineered ACRs for effective inhibition due to the prolonged open-state and the resulting accumulation of chloride (Cl$^-$) conducting channels during the illumination period.

In ChR2, the two amino acids C128 and D156 constitute the DC-gate, which is required for the reprotonation of the Schiff-base and subsequent termination of the conducting state. Mutations at these residues yielded slow-cycling, bistable CCRs, which could be toggled between a conducting and a closed state with light of different wavelengths[16–18]. Combining mutations at residues C128 and D156 in the same molecule resulted in particularly long-lasting conducting states[17]. To extend the temporal range for inhibition, slow-cycling ACRs with long-lasting conducting states were generated using a similar strategy[8,9,11,12,19]. Introduction of the analogous C128A mutation in engineered ACRs yielded the second-generation step-function ACRs SwiChR++[9] and Phobos$^{CA}$[11], with closing time constants of $115 \pm 9$ s and $249 \pm 10$ s, respectively. Channel closing of these ACRs could be strongly accelerated to less than 500 ms with orange light illumination, which allowed their use to reversibly inhibit various types of neurons in Drosophila larvae[11] and mice[9]. Similarly, cysteine-alanine substitution in the potent, natural GtACR1 (GtACR1$^{CA}$) has been shown to enable inhibition of neuronal spiking in mammalian neurons for at least 10 s after a brief light stimulus[20], as well as light-induced body wall muscle relaxation when expressed in motor neurons of C. elegans[21]. However, GtACR1$^{CA}$ is not fully switchable due to overlapping activation and inactivation spectra, limiting precise off-kinetics and temporal control of inhibition[20].

To further explore the possibilities of generating slow-cycling ACRs and to temporally extend their open state (i.e. the inhibition period), we systematically mutated amino acids in the DC-gate of Phobos$^{CA}$ and improved the trafficking properties in neurons. Of the various D156 mutations tested, Phobos$^{CADC}$ expressed well in neurons, showed enhanced photocurrents and a vastly extended life-time of the conducting state compared to its parental Phobos$^{CA}$. We therefore named this variant "Aion" after the Greek god of unbounded time[22]. Importantly, the photo-switching ability is preserved in Aion. It can be photoconverted to a fully closed state with light around 590 nm, granting precisely timed termination of silencing. We characterized the biophysical properties of Aion in HEK cells and show its silencing ability in organotypic hippocampal slice cultures, permitting faithful silencing of neurons over many hours with short light pulses spaced several minutes apart. We further show the utility of Aion in vivo by inhibiting nociceptive circuit function in Drosophila melanogaster larvae over extended time periods in all-optical experiments. Aion thus broadens the available toolkit of optogenetic silencers in the temporal domain, opening new possibilities for optical manipulation of neuronal circuits.

## Results

**Engineering of Aion and biophysical characterization in HEK cells.** Based on the previously engineered blue shifted, step-function ACR Phobos$^{CA}$[11], we systematically mutated the residue D156 (Fig. 1a), which together with C128 forms the DC-gate, a structural link between helices 3 and 4[23] that is important for channel gating kinetics in the parent ChR C1C2[24]. We exchanged D156 for either cysteine (C), asparagine (N), serine (S), or histidine (H)[25,26]. From all tested variants, only D156N (Phobos$^{CADN}$) and D156C (Phobos$^{CADC}$, i.e. Aion) yielded functionally expressed ACRs in neurons (Supplementary Fig. 1). We therefore characterized these variants in detail and compared them to the previously published slow-cycling ACRs Phobos$^{CA}$ and GtACR1$^{CA}$.

To assess the biophysical parameters of Aion and Phobos$^{CADN}$ and to benchmark them against Phobos$^{CA}$ and GtACR1$^{CA}$ we transiently expressed them in HEK-293 cells and measured light-induced photocurrents in voltage-clamp recordings (Fig. 1, Supplementary Fig. 2). Long lasting photocurrents were detected following brief illumination (20 ms) with blue or green light (Fig. 1b). Extending the illumination period to 500 ms revealed different modes of desensitization for all variants (decline of the peak current to stationary level, Supplementary Fig. 2a, b). We analyzed the peak ($I_p$) and stationary photocurrent 30 s post illumination ($I_{30s}$), normalized to cell capacity. Photocurrents of Phobos$^{CADN}$ were smaller compared to the parental Phobos$^{CA}$, while Aion displayed increased photocurrent densities (Fig. 1c). For the parental Phobos$^{CA}$, both currents were similar ($I_p = 39 \pm 4$ and $I_{30s} = 33 \pm 4$ pA/pF). The calculated desensitization ($15 \pm 1\%$) could be attributed to the slow channel closure as for Phobos$^{CA}$ no short-living initial photocurrent was observed (Supplementary Fig. 2b). In contrast, both, Phobos$^{CADN}$ and Aion displayed peak photocurrents that desensitized during the 500 ms illumination (Supplementary Fig. 2b), accounting for a $30 \pm 1\%$ and $38 \pm 2\%$ overall reduction of photocurrents after 30 s, respectively (Fig. 1c). For GtACR1$^{CA}$ desensitization was less pronounced during the 500 ms illumination period, while the photocurrent showed a biphasic decay with highly distinct time constants after light offset (Supplementary Fig. 2b), leading to a reduction by $74 \pm 2\%$ after 30 s. Notably, GtACR1$^{CA}$ peak photocurrent densities were highest ($105 \pm 20$ pA/pF) among all tested slow-cycling ACR

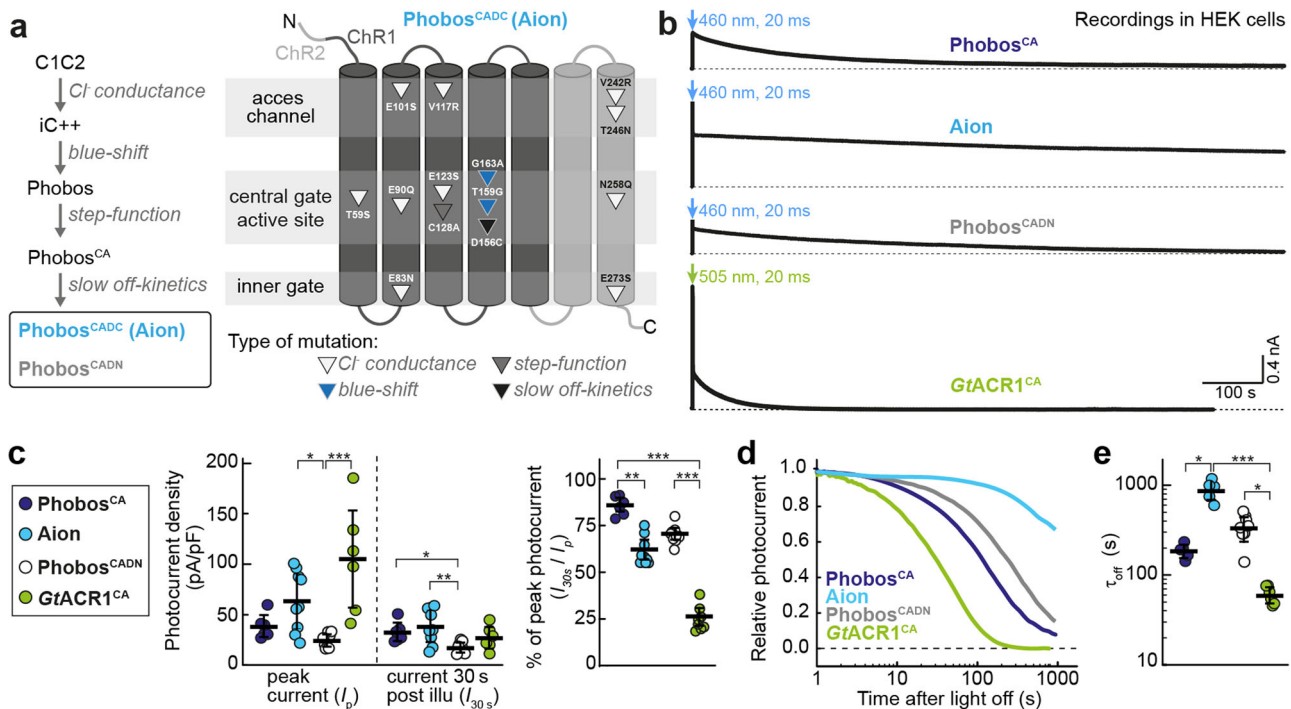

**Fig. 1 Development of Aion and Phobos^CADN and photocurrent characterization in HEK cells. a** Engineering strategy yielding Aion and Phobos^CADN, two new blue-shifted, step-function ACRs with temporally extended conducting states. Mutation of the D156 residue to either C or N (yielding Aion and Phobos^CADN, respectively) greatly slows down the closing kinetics of the ACRs compared to their parental construct Phobos^CA. The generated variants are highlighted by a black frame. The schematic drawing of Aion indicates point mutations (triangles) at the relative positions within the respective transmembrane helices. **b** Representative photocurrent traces of the slow-cycling ACRs Phobos^CA, Aion, Phobos^CADN, and GtACR1^CA activated by a short 20 ms light pulse at indicated wavelengths and irradiance of 3.35 mW/mm² (460 nm) and 3.14 mW/mm² (505 nm). **c** Quantification of peak photocurrent ($I_p$) and photocurrent 30 s post illumination ($I_{30s}$) for each ACR. The resulting photocurrent ratio at 30 s compared to the peak is plotted as percent of peak photocurrent. **d** Normalized photocurrent after light shutoff. **e** Closing time constants ($\tau_{off}$) for each ACR. For **c** and **e**, black lines correspond to mean values ± SEM, and circles are single measurement data points ($n_{PhobosCA}$ = 6 cells (**c**), 5 cells (**e**), $n_{Aion}$ = 9 cells (**c** left), 10 cells (**c** right), 5 cells (**e**), $n_{PhobosCADN}$ = 9 cells (**c** left), 11 cells (**c** right), 8 cells (**e**) $n_{GtACR1CA}$ = 6 cells (**c** left, **e**), 8 cells (**c** right)). Kruskal–Wallis test with Dunn's multiple comparisons test, *$p < 0.05$, **$p < 0.01$, ***$p < 0.001$.

variants (Fig. 1c). The closing time constants were $61 \pm 5$ s (GtACR1^CA), $186 \pm 14$ s (Phobos^CA), $338 \pm 36$ s (Phobos^CADN) and $892 \pm 82$ s (Aion, Fig. 1d, e) and could be readily accelerated with orange light for all Phobos-based variants, leading to complete termination of Cl⁻ conductivity (Supplementary Fig. 2a, c–e). In contrast, for GtACR1^CA only a small reduction of the conductance (to $70 \pm 2\%$ of the initial current) was achieved when red light[20] was applied 2 s after GtACR1^CA activation. Delaying red light application to 30 s caused reactivation and even increased photocurrent amplitudes (Supplementary Fig. 2a) due to the substantial overlap of conducting state and dark state.

In summary, Phobos^CADN and Aion displayed extended closing time constants compared to Phobos^CA. Aion showed the longest-lasting conducting state of all currently known ACRs, while preserving a large photocurrent density with full switchability between the open and closed state. Notably, introduction of the DC mutation led to slightly red-shifted peak activation and blue-shifted inactivation wavelengths ($\lambda_{activation} = 463.1 \pm 1.5$ nm; $\lambda_{inactivation} = 566.6 \pm 0.3$ nm; Supplementary Fig. 3b), compared to Phobos^CA ($\lambda_{activation} = 450$ nm; $\lambda_{inactivation} = 580$ nm; see ref. [11]) but the spectral separation is still large enough for full bidirectional switching. The shifts are likely due to the altered retinal binding pocket, as shown previously for the D156C mutation in ChR2[27]. However, the Aion modifications did not alter the reversal potential of the photocurrents ($E_{rev} = -63.8 \pm 0.8$ mV; Supplementary Fig. 3a) compared to the parental Phobos^CA ($-65.1 \pm 0.3$ mV, see ref. [11]) which means that anion selectivity was preserved in Aion.

**Functional characterization of Aion, Phobos^CADN, and GtACR1^CA in CA1 pyramidal neurons.** Next, we evaluated the functionality of Aion in comparison to Phobos^CADN and GtACR1^CA in CA1 pyramidal neurons of rat organotypic hippocampal slice cultures. All three citrine-labeled ACRs showed high expression and membrane-localized distribution (Fig. 2a). However, GtACR1^CA did not traffic evenly throughout the cell and displayed patchy and punctate accumulations along the neuronal arbors (Fig. 2a iii).

Using whole-cell patch-clamp recordings, we measured light-induced ACR-mediated Cl⁻ currents. Opsin-expressing neurons were held at a depolarized membrane voltage ($-35$ mV) with respect to the $E_{Cl^-}$. Photostimulation of ACRs thus results in outward-directed photocurrents upon Cl⁻ entry. A short light pulse at the peak wavelength (460 nm for Aion and Phobos^CADN; 525 nm for GtACR1^CA) elicited Cl⁻ photocurrents that were maintained for several seconds to minutes after light offset (Fig. 2b). While all ACRs showed similar peak current amplitudes (Fig. 2c), Phobos^CADN- and GtACR1^CA-mediated Cl⁻ photocurrents decayed faster than Aion-mediated currents. Peak-normalized mean photocurrent amplitudes dropped from $25.6 \pm 1.2\%$ at 30 s to $14.6 \pm 1.1\%$ at 120 s after photostimulation of Phobos^CADN (120 s / 30 s ratio: $59.4 \pm 10.1\%$). In GtACR1^CA-expressing cells stationary photocurrents were reduced from $20.2 \pm 1.1\%$ at 30 s to $6.1 \pm 1.8\%$ at 120 s after light stimulation (120 s / 30 s ratio: $29.5 \pm 8.2\%$) (Fig. 2d, e). In contrast, Aion-mediated stationary Cl⁻ currents were relatively stable over the whole 3-min recording (Fig. 2b i), with a 120 s / 30 s photocurrent

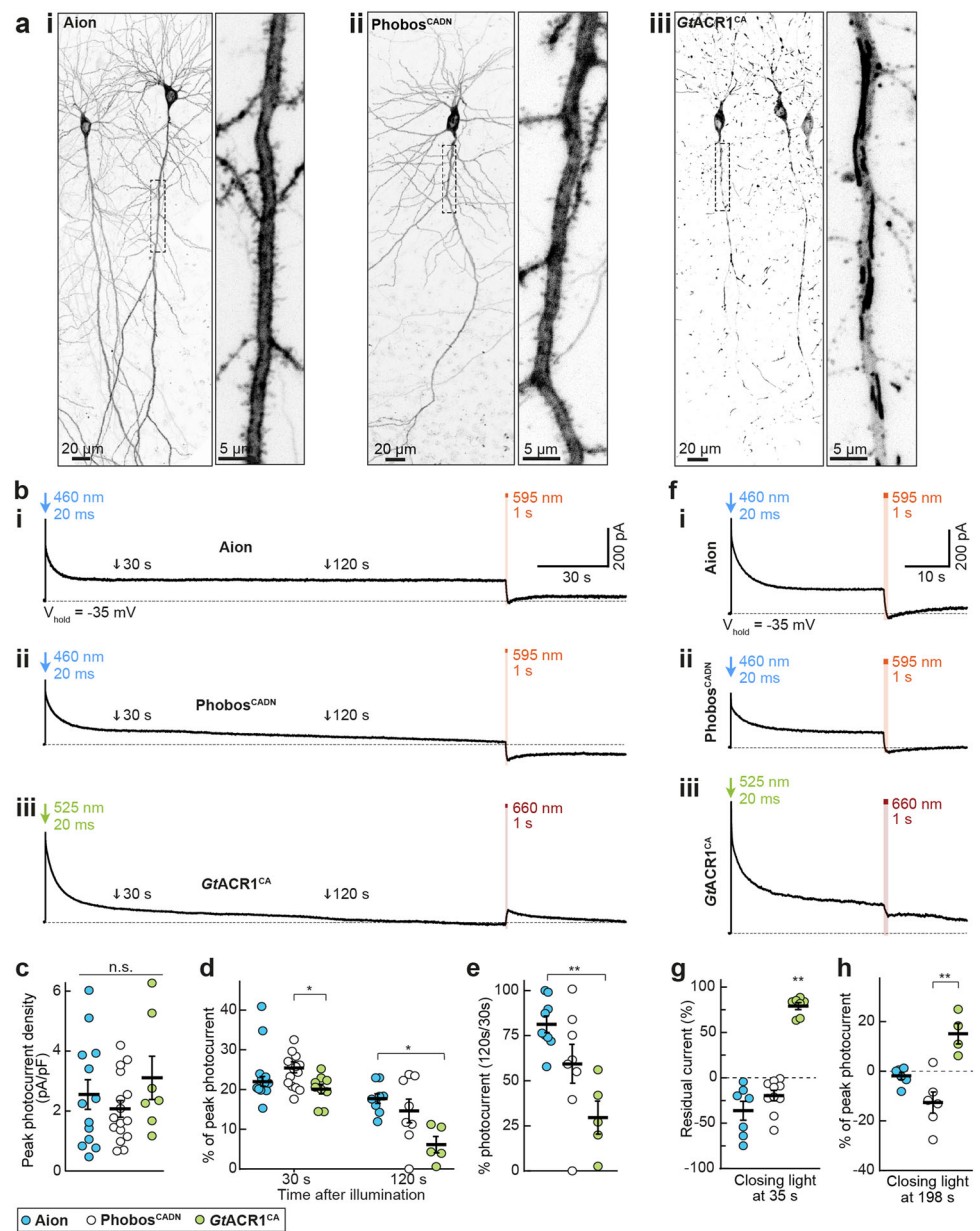

**Fig. 2 Expression and photocurrents of Aion, Phobos^CADN, and GtACR1^CA in CA1 pyramidal neurons. a** Maximum-intensity projections of two-photon image stacks showing expression of Aion (i) Phobos^CADN (ii) and GtACR1^CA (iii) in CA1 pyramidal neurons of rat organotypic hippocampal slice cultures. Fluorescence intensity is shown as inverted gray values. Insets show magnified view of the apical dendrite. **b** Representative photocurrent traces of Aion (i), Phobos^CADN (ii) and GtACR1^CA (iii), evoked by a 20 ms light pulse at the respective peak activation wavelength (460 nm for Aion and Phobos^CADN, 525 nm for GtACR1^CA) and equal irradiance of 10 mW/mm^2. Channel closing was accelerated with red-shifted light (1 s, 10 mW/mm^2, 595 nm for Aion and Phobos^CADN, 660 nm for GtACR1^CA) at 198 s after channel opening. Note that red light at 198 s elicited partial opening of GtACR1^CA instead of closing, indicated by an outward current after light stimulation. The black arrows at 30 and 120 s after channel opening indicate the time points at which stationary photocurrent amplitudes were quantified in **c–e**. **c** Quantification of peak photocurrent density for Aion, Phobos^CADN and GtACR1^CA. **d** Quantification of photocurrent amplitude at 30 s and 120 s after channel opening with respect to peak current. **e** Photocurrent ratio at 120 s compared to 30 s. **f** Same as (**b**) but channel closing was accelerated 35 s after opening. **g** Residual current upon channel closing 35 s after opening. Photocurrent amplitude was measured in the 5 s before and after the closing light pulse. **h** Quantification of photocurrent amplitude upon channel closing 198 s after opening with respect to the peak photocurrent. For (**c–e, g, h**), mean values ± SEM are shown (black lines) together with single measurement data points (circles, $n_{Aion} = 13$ cells (**c, d** left), 9 cells (**d** right, **e**), 7 cells (**g**), 6 cells (**h**); $n_{PhobosCADN} = 16$ cells (**c, d** left), 8 cells (**d** right, **e**), 10 cells (**g**), 6 cells (**h**); $n_{GtACR1CA} = 7$ cells (**c, g**), 9 cells (**d** left), 5 cells (**d** right, **e**), 4 cells (**h**)). Kruskal–Wallis test with Dunn's multiple comparisons test, *$p < 0.05$, **$p < 0.01$.

ratio of $81.3 \pm 4.3\%$ ($22.1 \pm 1.2\%$ and $17.7 \pm 1.1$ of peak photocurrent at 30 s and 120 s after photostimulation, respectively) (Fig. 2d, e).

Similar to the parental Phobos^CA [11], and consistent with the measurements in HEK cells (Supplementary Fig. 2), we could

accelerate the closing of both Aion and Phobos^CADN at any time point by illuminating them with orange light (595 nm). Absence of residual photocurrents after orange illumination suggests full channel closing (Fig. 2b i and ii, f i and ii, g, h). However, this was not the case for GtACR1^CA, for which 1 s illumination with red

light (660 nm) at 35 s after channel opening reduced the $Cl^-$ current only by 21% (Fig. 2f iii, g). Moreover, red light stimulation 198 s after channel opening, a time by which $Gt$ACR1$^{CA}$ was mostly closed via thermal relaxation of the open state, elicited a $Cl^-$ current with an amplitude of $17.2 \pm 3.6\%$ compared to the green-light induced peak current (Fig. 2b iii, h). This suggests that red light likely brings $Gt$ACR1$^{CA}$ to an intermediate conducting state in the photocycle due to a residual absorption of the dark state at 660 nm.

Notably, a large, sustained $Cl^-$ conductance in depolarized neurons under voltage-clamp may slowly alter $Cl^-$ homeostasis, leading to a shift in the $Cl^-$ reversal potential and activation of cation-chloride co-transporters such as KCC2[28]. This makes the precise quantification of reversibility of ACR-activation challenging in neurons. Thus, we turned to a more natural configuration and recorded neurons under current clamp, near their resting membrane voltage and their natural $Cl^-$ Nernst potential. We asked whether Aion and Phobos$^{CADN}$ are suitable to block action potentials (APs) after a short light pulse and whether this block could be reverted with red-shifted illumination, yielding a similar spike rate as before the inhibition (Fig. 3, Supplementary Fig. 4). Both Aion and Phobos$^{CADN}$ efficiently blocked depolarization-induced APs during a 40 s period following a 20 ms blue light pulse. Furthermore, we could recover AP firing with high temporal precision after closing the respective ACR with orange light (Fig. 3a, Supplementary Fig. 4a). While activation of $Gt$ACR1$^{CA}$ (20 ms, 525nm-light pulse) also resulted in sustained inhibition of current-evoked APs, firing could not be restored upon red illumination (Fig. 3b). This is consistent with the photocurrent recordings in neurons and HEK cells, which showed only partial reduction of $Gt$ACR1$^{CA}$ conductance upon red illumination (Fig. 2f iii, g; Supplementary Fig. 2a, d). Due to the residual $Cl^-$ conductance after illumination with red light, APs did not recover during the 55 s recording period.

To compare the silencing performance of the three step-function ACRs more quantitatively, we measured their capability to positively shift the rheobase (i.e. the minimum current required to elicit an AP). We repeatedly injected depolarizing current ramps into ACR-expressing CA1 neurons at an interval of 0.2 Hz for 1 min (see Methods section for further details). For each ramp, the injected current at the time of the first AP was defined as the rheobase. To open the respective ACR a 1 s light pulse was applied after the first current ramp (Fig. 3c, g; Supplementary Fig. 4b). All three ACRs shifted the rheobase towards larger currents for several seconds after light stimulation, albeit with different efficacy. Phobos$^{CADN}$ showed the weakest silencing capacity, only shifting the rheobase by $157.5 \pm 32.7$ pA immediately after illumination, which was not sufficient to completely block APs (Supplementary Fig. 4c, d). Aion-mediated inhibition was more potent, yielding a rheobase shift of $463.2 \pm 98.0$ pA which was maintained throughout the 47-s recording period after light stimulation (Fig. 3d). As a result, the median number of current-evoked APs was suppressed by 85.3% (range: 80.0–93.3%), with complete AP block in 2 out of 7 cells (Fig. 3e). $Gt$ACR1$^{CA}$ showed the strongest inhibition immediately after light stimulation, shifting the rheobase by $601.3 \pm 162.9$ pA and fully suppressing APs in 4 out of 6 cells. However, as expected from the faster closing time constant, $Gt$ACR1$^{CA}$ silencing capacity was not sustained over the time course of 47 s. The shift in rheobase dropped to $318.2 \pm 142.6$ pA (a reduction of 52.9%), yielding 56.2% (range: 0.0–100%) suppression of APs (Fig. 3h, i). All three ACRs significantly decreased AP firing starting at an irradiance of 0.1 mW/mm$^2$, with $Gt$ACR1$^{CA}$ showing complete block of APs already at 0.01 mW/mm$^2$ in 2 out of 7 cells (Fig. 3f, j; Fig. 3e). These results match the photocurrent recordings (Fig. 2b–e) and indicate that,

while $Gt$ACR1$^{CA}$ shows more potent inhibition immediately after light offset, Aion shows the best combination of acute and sustained inhibition of neuronal spiking. Given the weak light-mediated AP suppression in Phobos$^{CADN}$-expressing neurons (Supplementary Fig. 4), we only further assessed Aion and $Gt$ACR1$^{CA}$ for their applicability as long-term silencing tools.

So far, we evaluated silencing of APs over a relatively short time period, which was already achievable with previously published slow-cycling ACRs. We now aimed to test neuronal silencing over time periods of many minutes and up to 12 h. First, we quantified the effects of Aion and $Gt$ACR1$^{CA}$ to assess their suitability for long-term silencing experiments. In this context, Aion reliably blocked depolarization-induced APs over a 10-min period when activated by 2 brief light pulses (1 s, 460 nm, 10 mW/ mm$^2$) delivered 5 min apart (Fig. 4a). Accordingly, the Aion-mediated decrease in membrane depolarization of 54.5% (range: 48.2–62.4%) was stable for 5 min after light stimulation and was sufficient to block all APs (Fig. 4b, c). In contrast, in neurons expressing $Gt$ACR1$^{CA}$ current-evoked AP firing was generally recovered within the first minute (median 58.8 s, range: 20.5–201.6 s) after opening of the ACR by a short light pulse (1 s, 525 nm, 10 mW/mm$^2$), and the initial reduction of 59.3% (range: 52.7–66.9%) in membrane depolarization went back to baseline values after $3.6 \pm 0.3$ min (Fig. 4d–f). Therefore, reliable AP suppression with $Gt$ACR1$^{CA}$ for time periods longer than 1 min requires light stimulation protocols with shorter inter-stimulus intervals, and consequently more frequent light exposure, as compared to Aion.

Given that brief light stimulations every 5 min allowed reliable suppression of AP firing in Aion-expressing CA1 neurons, we next asked whether Aion is suitable for neuronal silencing over several hours using 5-min long dark interstimulus intervals. Neurons were stimulated overnight (O/N) inside the incubator in a custom-made blue-LED chamber (Fig. 4g). After 12 h of pulsed blue stimulation, we transferred the slices to the patch-clamp setup and did whole-cell current-clamp recordings in Aion-expressing cells to assess whether baseline spiking performance was unaltered and whether Aion-mediated silencing was still functional. We ensured that residual Aion conductance was terminated by applying orange light before the start of the recordings. As before, APs were efficiently evoked by 2-s current injections and reliably blocked for at least 5 min after activation of Aion with a single blue light pulse (3 s, 0.3 mW/mm$^2$, same conditions as O/N stimulation in the incubator, Fig. 4g, h). Thus, Aion was still fully functional and silencing was efficient after O/N activation in the incubator.

To determine whether such long-term Aion-mediated $Cl^-$ conductance compromised cell health, we measured passive and active membrane parameters of Aion-expressing neurons after the O/N protocol. Overall, no significant differences were observed compared to three different control conditions: non-transfected or citrine-expressing CA1 cells that received O/N photostimulation, or Aion-expressing neurons that were not illuminated (Fig. 4i). Only the resting membrane potential was mildly depolarized in citrine-only (but not Aion) cells and there was a trend towards lower membrane resistance in Aion expressing cells independent of the illumination. This suggests that activation of Aion for 12 h under the indicated conditions was well tolerated by CA1 pyramidal neurons and did not lead to obvious alterations of $Cl^-$ homeostasis.

**Improving and restricting membrane expression of Aion to the somatodendritic compartment.** For the functional characterization of Aion in CA1 pyramidal neurons, we used a final plasmid concentration of 20 ng/μl for single-cell electroporation.

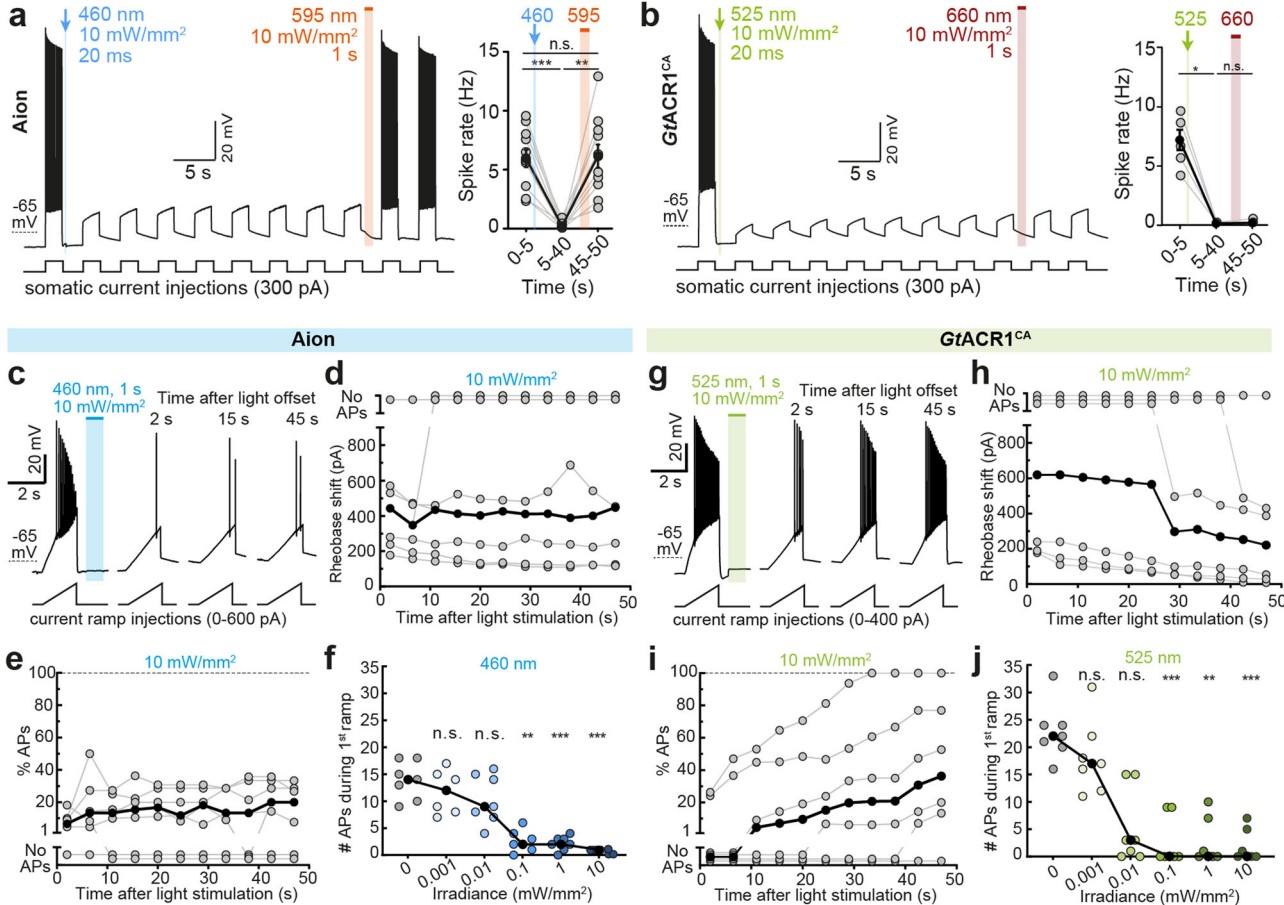

**Fig. 3 Silencing efficiency of Aion and GtACR1^CA in CA1 pyramidal neurons. a** Membrane voltage trace showing reversible suppression of depolarization-induced action potentials (APs) by photoswitching Aion between open and closed state with a blue (460 nm, 20 ms, 10 mW/mm²) and an orange light pulse (595 nm, 1 s, 10 mW/mm²), respectively. Right: Quantification of spike rate during current injection at indicated time intervals: before channel opening (0–5 s), after channel opening (5–40 s), and after channel closing (45–50 s) in Aion-expressing CA1 neurons (n = 10 cells). **b** Same as (**a**) but for neurons expressing GtACR1^CA (n = 5 cells). The channel was opened with a green (525 nm, 20 ms, 10 mW/mm²) light pulse. Note that in GtACR1^CA-expressing cells AP firing could not be recovered immediately after illumination with red light (660 nm,1 s, 10 mW/mm²). **c** Current ramps were injected into Aion-expressing neurons to induce APs before and after illumination with a short blue light pulse (460 nm, 1 s, irradiances from 0.001 to 10 mW/mm²). For each ramp, the injected current at the time of the first AP was defined as the rheobase. Example membrane voltage traces are shown for the trial in which a light intensity of 10 mW/mm² was used. **d** Quantification of the rheobase shift and (**e**) the relative change in the number of current ramp-evoked APs over 47 s after light stimulation (460 nm, 1 s, 10 mW/mm²) (n = 7 cells). **f** Number of APs evoked during the first current ramp after opening of Aion with 1 s blue light at indicated irradiances. Significant AP block was achieved at 0.1 mW/mm² (n = 7 cells). **g–j** Same experiment as shown in (**c–f**) except that CA1 neurons expressed GtACR1^CA (n = 6 cells in **h**, **i** and 7 cells in **j**) and 525 nm-light was used for channel opening. Note that GtACR1^CA silencing efficacy decayed already within the first minute after light stimulation (**i**, **j**). For (**a**, **b**, **d–f**, and **h–j**) filled circles represent single measurement data points and black circles correspond to medians, Friedman test, *p < 0.05, **p < 0.01, ***p < 0.001, n.s. not significant.

This concentration led to robust, membrane-localized expression (Fig. 2a i) and was well tolerated by the cells, leaving basic neuronal parameters intact (Fig. 4i). In some cases, opsin accumulations were visible at the soma, the basal dendrites and the proximal region of the main apical dendrite (Supplementary Fig. 5a). Notably, despite these local accumulations, there was still significant membrane expression of Aion detectable throughout the cell (Supplementary Fig. 5a i). Nonetheless, to improve protein trafficking and to avoid local protein accumulations, we added the plasma membrane trafficking signal (ts: KSRITSE-GEYIPLDQIDINV) from the mammalian inward rectifying K⁺ channel Kir2.1 between the coding sequences for Aion and citrine and the Kir2.1 endoplasmic reticulum export signal (ER: FCYE-NEV) at the C-terminus of citrine. Both modifications were previously shown to reduce intracellular aggregation of the ACR GtACR2[29] and the Cl⁻ pump NpHR[30]. The resulting construct,

Aion-ts-citrine-ER, displayed enhanced membrane trafficking with no detectable intracellular accumulations (Supplementary Fig. 5b). Moreover, Aion-ts-citrine-ER showed similar silencing capacity as the original Aion when electroporated at a fourfold lower concentration (5 vs. 20 ng/µl), yielding a sustained rheobase shift of 404.5 ± 111.3 pA and efficient suppression of APs (82.6%, range: 67.5–100%) during the entire 47 s recording period after stimulation with a brief blue light pulse (Supplementary Fig. 5c–f; see Fig. 3c–f for original Aion).

We used a second approach to improve membrane trafficking of Aion by attaching a C-terminal Kv2.1-trafficking sequence, which in addition of enhancing membrane localization, restricts expression to the somatodendritic compartment[31]. Soma targeting has the added benefit of preventing expression of the ACR in axon terminals where functionality might be limited due to an excitatory $E_{Cl^-}$[29,32]. The soma-targeted variant of Aion (somAion) showed enhanced

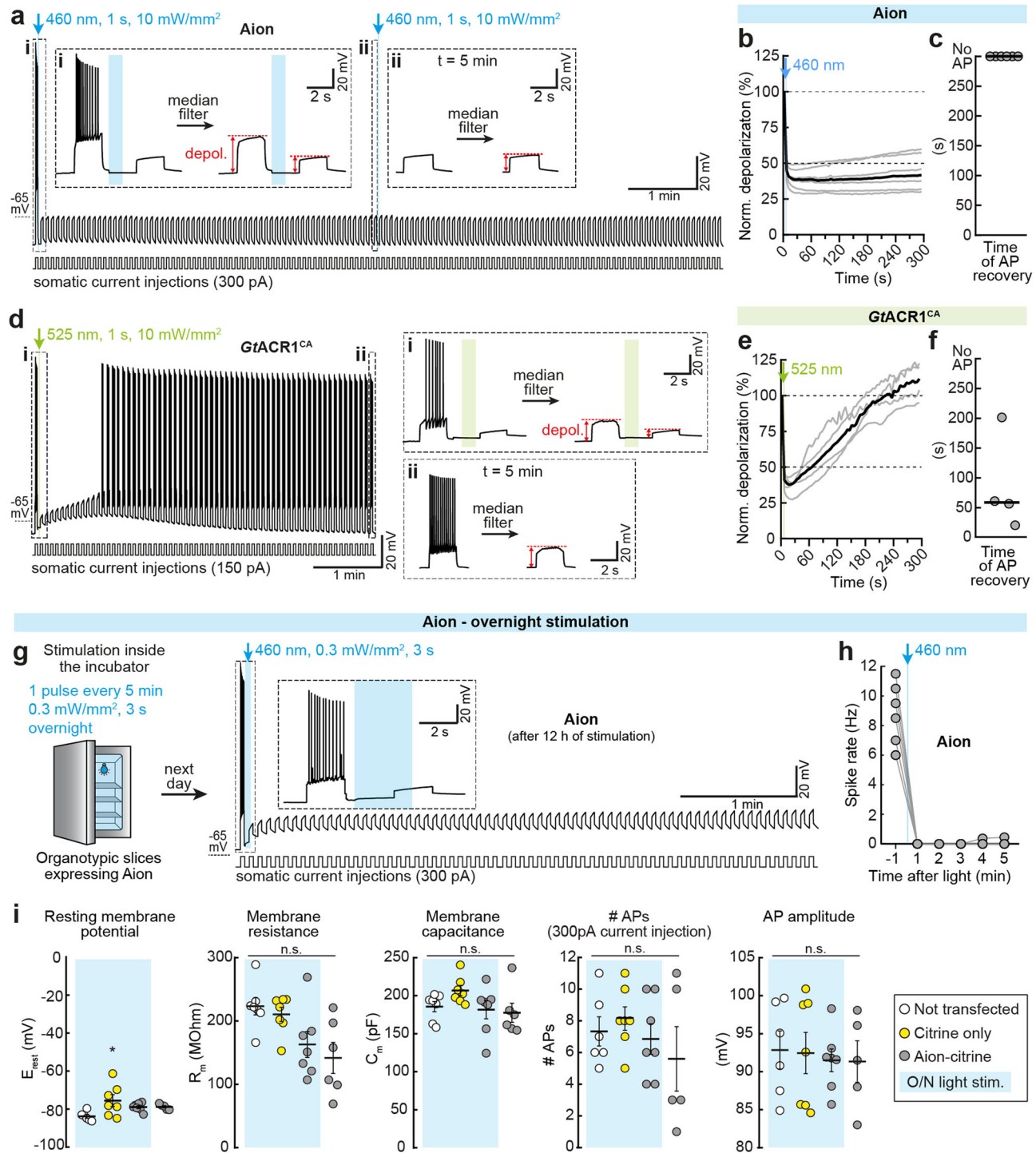

membrane trafficking and was confined to the soma and main apical dendrite (Fig. 5a, b). Despite local restriction of somAion its photocurrent densities were slightly enhanced (peak: $1.5 \pm 0.3$ pA/pF; stationary: $0.4 \pm 0.1$ pA/pF) compared to those in neurons expressing the non-soma-targeted Aion (peak: $1.3 \pm 0.3$ pA/pF; stationary: $0.3 \pm 0.1$ pA/pF) (Fig. 5c). Furthermore, consistent with the efficient membrane expression in the somatodendritic compartment, somAion yielded stronger inhibition of AP firing than non-soma-targeted Aion, shifting the rheobase to higher values and leading to a complete and sustained block of current-evoked APs in most cells (4 out of 6 cells, Fig. 5d–f, see Fig. 3c–f and Supplementary Fig. 4d, e for comparison to non-soma-targeted Aion versions). Taken together, soma-targeting of Aion overcomes the suboptimal trafficking, while

avoiding expression in axon terminals, yielding a more effective tool for optogenetic inhibition at the cell soma.

**All-optical manipulation of a nociceptive circuit in *Drosophila* larvae.** To assess the functionality of Aion in vivo, we analyzed its ability to inhibit nociceptive circuit function in *Drosophila* larvae. We expressed Aion in a pair of major second order neurons (A08n) downstream of the primary nociceptors (C4da) covering the larval body wall (Fig. 6a, b). C4da and A08n neurons are part of a circuit sensing noxious mechanical stimuli, and their activation elicits stereotyped rolling behavior[33,34]. After activation of Aion in A08n neurons with a blue light pulse, we observed a

**Fig. 4 Long-term silencing with Aion and *Gt*ACR1[CA] in CA1 pyramidal neurons. a** Aion silencing capacity was evaluated over 10 min. Example membrane voltage trace of an Aion-expressing CA1 neuron showing reliable suppression of depolarization-induced APs (2 s current injections every 4.5 s) for 10 min by activating Aion with 2 short blue light pulses spaced 5 min apart (460 nm, 1 s, 10 mW/mm$^2$). (i, ii) Insets show magnified view of the membrane potential at the indicated time points. Voltage traces were median-filtered to calculate the change in membrane depolarization after light stimulation of Aion (as shown by the red arrows). **b** Quantification of Aion-mediated change in membrane depolarization over a time period of 5 min after light stimulation (460 nm, 1 s, 10 mW/mm$^2$) as shown in (**a**). Gray traces correspond to single neurons and black trace shows the median ($n = 6$ cells). **c** Time at which the first current-evoked AP occurred after opening of Aion. Note that Aion reliably blocked all APs for 5 min after light stimulation. Gray circles represent single measurement data points and black line corresponds to median ($n = 6$ cells). **d** *Gt*ACR1[CA] silencing capacity was evaluated over 5 min. Example membrane voltage trace of a *Gt*ACR1[CA]-expressing CA1 neuron after activation of *Gt*ACR1[CA] with a short green light pulse (525 nm, 1 s, 10 mW/mm$^2$). (i, ii) same as (**a**, i–ii) but for *Gt*ACR1[CA]. **e, f** Same as (**b**, **c**) but for *Gt*ACR1[CA] ($n = 4$ cells; 525 nm, 1 s, 10 mW/mm$^2$). Note that the change in membrane depolarization reverted to baseline approx. 3.5 min after light stimulation (**e**) and neurons started firing within the first 200 s after light stimulation (**f**). **g** Organotypic slices with CA1 cells expressing Aion were stimulated overnight (O/N) in a custom-made LED chamber inside the incubator (3 s 460-nm light pulse every 5 min, 0.3 mW/mm$^2$). Example membrane voltage trace of an Aion-expressing CA1 neuron recorded after 12 h of light stimulation in the incubator. Inset shows magnified view of the membrane potential during light stimulation. **h** Quantification of spike rate before and during the 5 min following light stimulation ($n = 6$ cells). **i** Resting membrane potential, membrane resistance, membrane capacitance, number of APs evoked by somatic current injection (300 pA, 500 ms) and amplitude of the first AP in Aion-expressing cells after 12 h of light stimulation, compared to the following three control groups: Non-transfected CA1 pyramidal cells, cells expressing only the fluorescent protein citrine that were stimulated O/N under the same conditions as Aion-expressing cells and Aion-expressing cells without O/N stimulation. Black lines: mean values ± SEM, $n_{\text{Non-transfected}} = 6$ cells ($E_{\text{rest}}$, #APs, AP amplitude), 7 cells ($R_m$, $C_m$); $n_{\text{citrine-only}} = 7$ cells; $n_{\text{Aion}} = 7$ cells; $n_{\text{Aion No light}} = 5$ cells ($E_{\text{rest}}$, #APs, AP amplitude), 6 cells ($R_m$, $C_m$), Kruskal–Wallis test with Dunn's multiple comparisons test, *$p < 0.05$, n.s. not significant.

robust reduction in nociceptive touch responses, which could be completely reverted after orange light application to accelerate channel closure of Aion (Fig. 6b). The effect was comparable to A08n inactivation with other silencing tools, including the potent natural *Gt*ACR1[35] upon acute green light exposure or chronic silencing with the inward rectifying potassium channel Kir2.1 (Fig. 6c, d). We next tested Aion-mediated silencing in an all-optical approach: we expressed the red-shifted ChR *Cs*Chrimson specifically in the upstream C4da nociceptors and Aion in A08n neurons in the same larvae. *Cs*Chrimson-expressing C4da neurons were activated with red light before and after Aion activation, as well as after Aion inactivation. In this case we capitalized on the slow off-kinetics of Aion using 100x lower light intensity for its activation over a prolonged period of 30 s to avoid blue-light-mediated cross-activation of *Cs*Chrimson. This light protocol resulted in comparable inhibition of behavior as a shorter light pulse with higher intensity (Fig. 6e, f). In both cases, *Cs*Chrimson-induced nociceptive behavior was strongly reduced after Aion activation and could be largely recovered upon Aion inactivation with orange light (Fig. 6e). Without subsequent Aion inactivation, nociceptive behavior remained strongly reduced after 5 and 15 min, and only partially recovered at 60 min after Aion-induced silencing of A08n neurons, demonstrating the long-lasting silencing capacity of this opsin (Fig. 6f).

We further generated transgenic flies with Aion variants optimized for expression (Aion-ts-mScarlet-ER) or somatodendritic targeting (somAion) as described above (see Supplementary Fig. 5 and Fig. 5) to investigate if these modifications might be beneficial for Aion functionality in *Drosophila* as well. These Aion variants placed in transgene backbones were tested for high or low expression (20xUAS or 5xUAS) in our all-optical paradigm as before (see Fig. 6e and Supplementary Fig. 6a). Both Aion variants were able to significantly inhibit nociceptive behavior after a low or high intensity blue light stimulus (Supplementary Fig. 6b–d). However, Kv2.1 inclusion did not improve the potency of Aion — likely due to inefficiency of this signaling sequence in *Drosophila*. Moreover, high expression of the other trafficking-enhanced Aion constructs had detrimental effects on baseline behavior before A08n silencing. Nonetheless, as with CA1 pyramidal neurons, lower expression of trafficking-enhanced Aion was similarly sufficient to inhibit nociceptive behavior suggesting that tuning Aion levels might be beneficial to optimize behavioral experiments in different neuronal subtypes.

Overall, these results show that Aion can robustly and reversibly inhibit behavior by light-induced silencing of specific neurons in vivo, as well as maintain light-independent silencing for extended periods of time, allowing manipulation of animal behavior without continuous illumination.

## Discussion

Here we sought to generate a potent, slow-cycling bimodally switchable ACR with a temporally extended conducting state by combining the C128A mutation in Phobos[CA][11] with a number of D156 mutations. The amino acids C128 and D156 form the DC-gate and are crucial for the reprotonation of the Schiff base, which terminates the conducting state[18,36,37]. In CCRs these residues were systematically mutated to obtain stabilized, step-function opsins with open state time constants of up to 30 min[17,38]. Mutagenesis at either of the two residues (C128, D156) was previously shown to also extend the conducting state of various engineered and natural ACRs[8,9,11,12,19,20]. However, the channel-closing time constant of published slow-cycling ACRs was so far not exceeding 5 min[9,11,19]. Thus, we sought to combine mutations at both residues, while preserving channel functionality and anion selectivity. This effort led to Aion, a step-function ACR with enhanced potency and a channel-closing time constant of approximately 15 min (Fig. 7).

Due to its slow photocycle, Aion yields effective inhibition with short light pulses delivered several minutes apart (Fig. 4) or with prolonged, low-irradiance illumination (Fig. 6e, f), thereby reducing undesired side effects of high photon flux over long time periods in optogenetic silencing experiments. These side effects include tissue heating, phototoxicity or disturbance of the sensory system (perception of light) during behavioral experiments. Furthermore, since the slow photocycle enables accumulation of conducting (i.e. open) anion channels under low photon exposure over time, Aion is in principle also suitable for long-lasting silencing of deep brain regions where light penetrance is low and capture of photons by the retinal is unlikely. At the same time, this might pose a limitation in experiments, where spatially confined silencing is required, since Aion may get activated outside the target area under continuous illumination, even at low irradiances far away from the light source[39,40].

Light-gated K[+]-channels with long-lasting action might pose a tractable alternative to ACRs. Activation of a K[+] conductance should lead to robust hyperpolarization and membrane shunting independently of the developmental state or the cellular

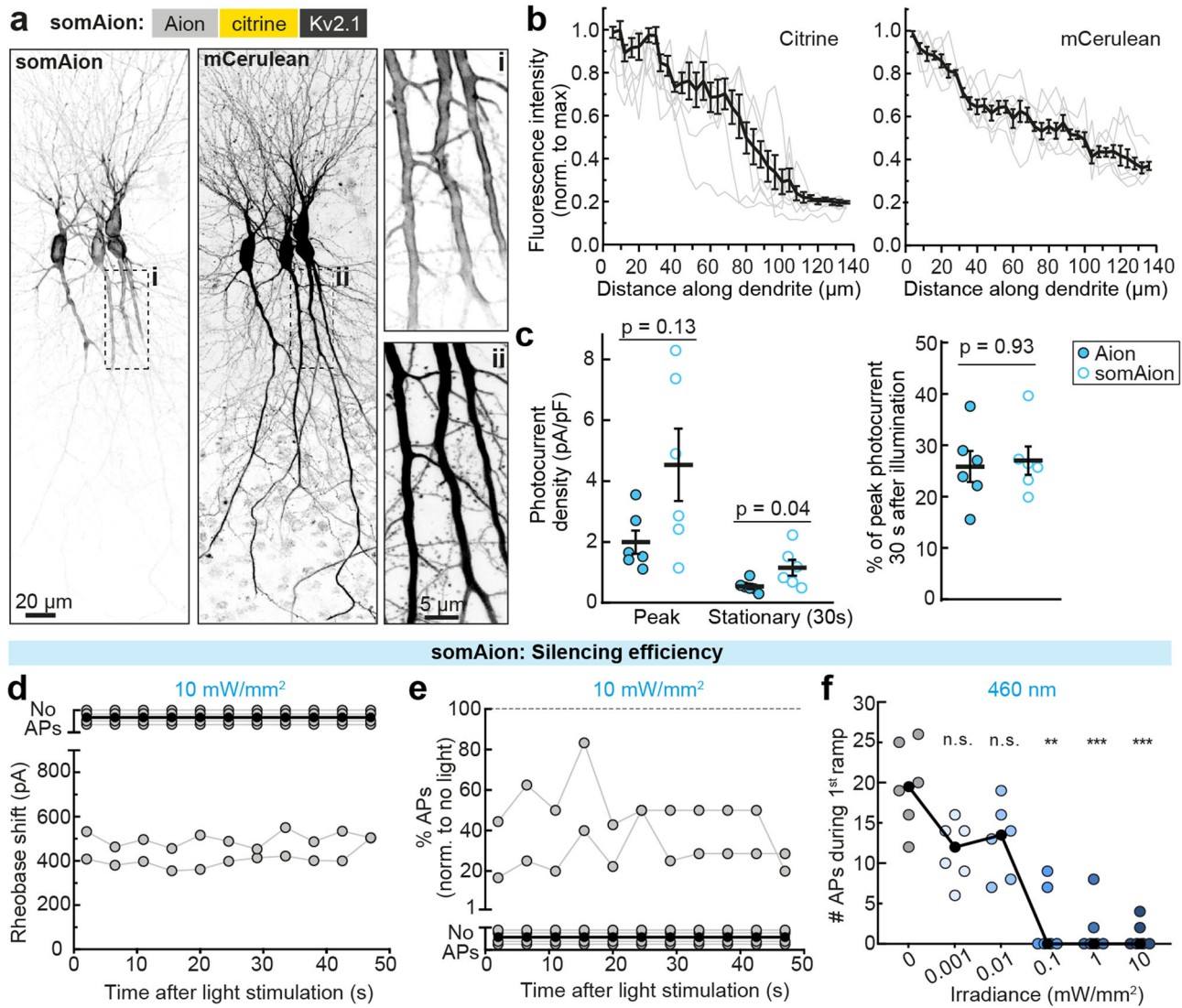

**Fig. 5 somAion is a soma-targeted variant of Aion with enhanced silencing capacity. a** Maximum-intensity projections of two-photon image stacks showing expression of somAion in CA1 pyramidal neurons after single-cell electroporation in organotypic hippocampal slice cultures. Fluorescence intensity is shown as inverted gray values. Opsin-citrine fluorescence was localized at the plasma membrane and restricted to the somatodendritic compartment (left image, inset i). mCerulean serves as a morphology marker (right image, inset ii). **b** Quantification of citrine and mCerulean fluorescence along the main apical dendrite of somAion-expressing neurons ($n = 7$ cells). Gray lines correspond to measurements in individual cells. Black lines indicate mean ± SEM. **c** Comparison of Aion and somAion peak and stationary (30 s post light) photocurrent densities evoked by a short 460-nm light pulse (1 s, 10 mW/mm²). Right: quantification of stationary, peak-normalized photocurrent amplitude at 30 s after channel opening. Black lines indicate mean ± SEM and circles represent measurement from single cells ($n_{Aion} = 6$ cells, $n_{somAion} = 6$ cells). Mann–Whitney test, exact $p$-values are shown. **d**–**f** Rheobase measurements as described previously. **d** Quantification of the rheobase shift and (**e**) the relative change in the number of current ramp-evoked APs during 47 s after light stimulation (460 nm, 1 s, 10 mW/mm², $n = 6$ cells). In most cells, activation of somAion led to complete AP block. **f** Number of APs evoked during the first current ramp after opening of somAion with 1 s blue light at indicated irradiances. Significant AP block was achieved at 0.1 mW/mm² ($n = 6$ cells). For **d**–**f** gray or blue circles represent single measurement data points and black filled circles correspond to medians, Friedman test, *$p < 0.05$, **$p < 0.01$, ***$p < 0.001$, n.s. not significant.

compartment of the neuron. Recently, two-component silencing tools were reported that employ cyclic nucleotide-gated K⁺-channels in combination with blue light-activated nucleotide cyclases[41,42]. Indeed, potent silencing of neuronal activity was demonstrated with these tools, including silencing of neuronal activity in the mammalian brain in vivo. While providing a promising new direction for efficient, light-sensitive silencing, putative caveats are off-target effects due to light-induced elevation of cyclic nucleotides and background-activation of the cyclic nucleotide-gated K⁺-channels by endogenous cyclic nucleotides. Such side effects may explain the unexpected alterations of hippocampal activity in mice expressing these tools[43]. The recently discovered natural K⁺ channels[44] might

provide a solution to such limitations. However, in all CCRs the main conducting photocycle intermediate contains a protonated retinal Schiff base with an unavoidable spectral overlap of dark and conducting states, which makes clean off-switching incomplete, which we consider as a mayor limitation of step-function CCRs.

Efficient optogenetic inhibition in deep brain areas was also achieved by using the red-light-activated Cl⁻ pump Jaws[45]. This study demonstrated inhibition of neurons in the mouse cortex over a range of 1–3 mm through the intact skull. However, as previously discussed, ion pumps present some limitations, such as actively changing the intracellular ionic composition, especially when prolonged inhibition periods are required[1,6,7]. In this

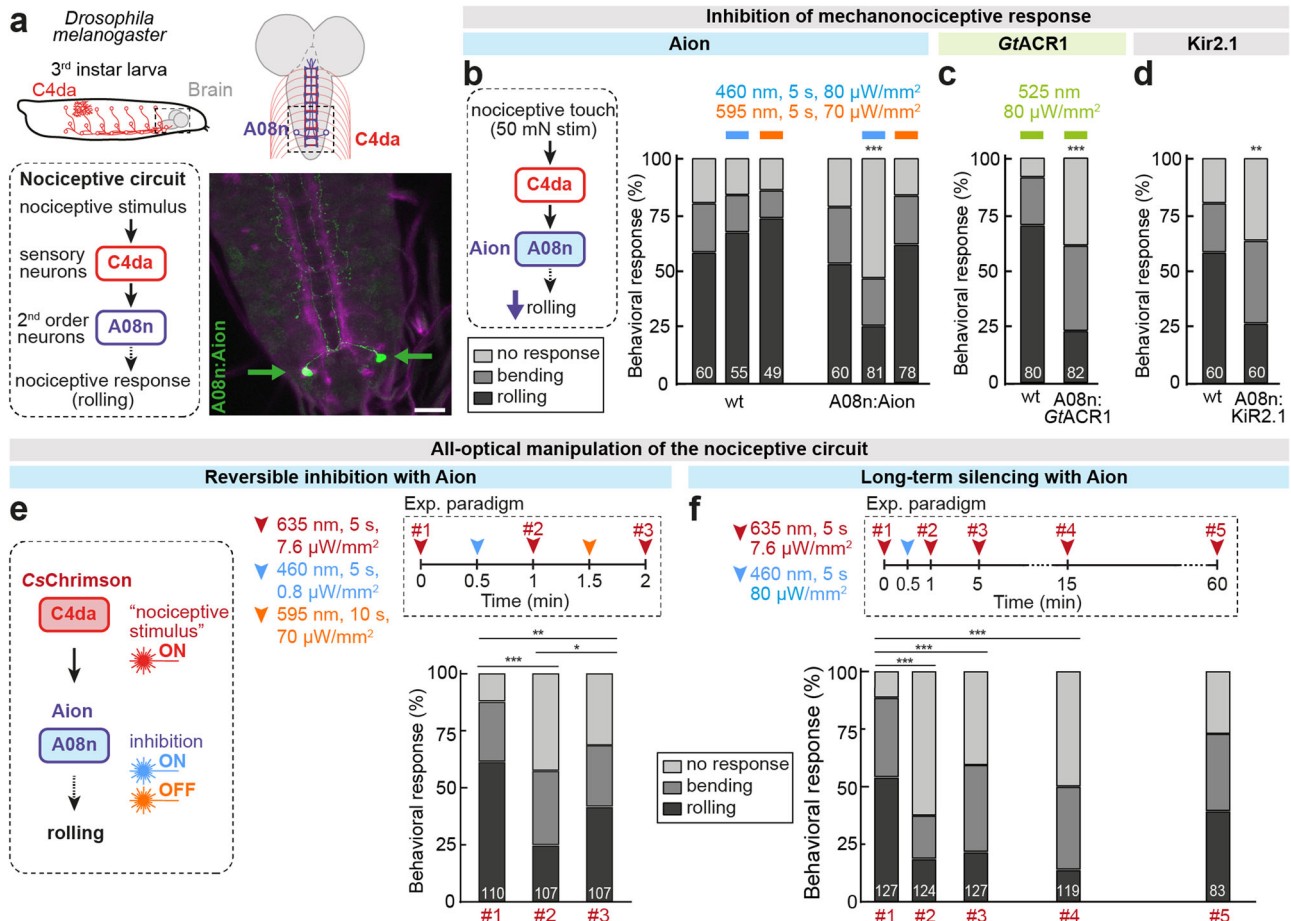

**Fig. 6 Aion enables inhibition of nociceptive circuit function in *Drosophila melanogaster* larvae over extended time periods. a** Schematic model of nociceptive sensory neurons (C4da) and connected downstream neurons (A08n) in *Drosophila* larvae. The example image shows Aion expression in A08n neurons (green, maximum intensity projection of confocal image stack). Fas3 staining (magenta) was used as a sensory axon marker for orientation. Arrows indicate cell bodies of A08n neurons. Scale bar: 20 µm. **b** Nociceptive touch responses were strongly reduced after blue light in animals expressing Aion in A08n neurons, but not in controls (wt). The responses could be fully recovered after orange light exposure. **c** Larvae expressing *Gt*ACR1 in A08n neurons showed robustly reduced nociceptive responses under constant green light exposure. **d** Constitutive silencing by expression of Kir2.1 in A08n neurons leads to similarly strong inhibition of larval nociception as with Aion. For **b–d** the behavioral response to a 50 mN mechanical stimulus is shown. **e** All-optical paradigm for inhibition of A08n neurons with Aion and activation of C4da neurons with *Cs*Chrimson. Repeated induction of nociceptive behavior with red light was strongly inhibited after blue light-induced Aion activation and largely reversed by orange light. **f** Activation of Aion expressed in A08n neurons with a single blue light pulse (5 s) resulted in reduction of nociceptive responses for at least 15 min and partially recovered only after 60 min. For **e**, **f** behavioral responses to *Cs*Chrimson activation in C4da neurons at the indicated time points are shown. $X^2$-test, n of animals are indicated by white numbers in **b–f**. For clarity, only significant differences are indicated. *$p < 0.05$, **$p < 0.01$, ***$p < 0.001$ For further details on statistics see Supplementary Data 1.

context, minimally invasive optogenetic inhibition with slow-cycling ACRs such as Aion could hold the advantage that ionic gradients are not actively changed, as they operate by shunting rather than hyperpolarization of the membrane voltage.

Another caveat of optogenetic silencers that require continuous illumination is the potential interference with behavior during in vivo experiments, which can ultimately lead to misinterpretations regarding the function of the neural circuit under investigation. Many animal species, including *C. elegans* or *Drosophila* larvae, detect and avoid short-wavelength light in the UV-blue spectrum even at low intensities[46–49]. This makes the implementation of optogenetic manipulations challenging in living animals, especially when illumination of the entire body is required. As shown previously for Phobos[CA 11], expressing Aion in *Drosophila* larvae we could temporally dissociate activation of the ACR from the behavioral manipulation (i.e. mechanical stimulus). Therefore, Aion allows temporal dissociation of

functional silencing and the physiological responses evoked by the light stimulus directly.

Step-function ACRs further allow optogenetic silencing together with temporally precise optical activation of a second population of neurons in vivo, despite some spectral overlap of both opsins. Even the most red-shifted opsins show significant light absorbance in the blue range. Therefore, combining a red-shifted excitatory opsin, such as Chrimson[50] in one neuronal population, with blue light-sensitive inhibitory opsins in a second neuronal population is complicated by the putative blue light-activation of Chrimson during activation of the silencing tool. One way to address this issue can be by temporally dissociating the activation of the two opsins. Switching Aion to an open state with blue light before the start of the manipulation of the second population of Chrimson-expressing neurons allows to circumvent unintended optical cross-talk, as shown in Fig. 6e, f. Another way to avoid blue-light-mediated activation of Chrimson during the activation of the ACR is to use prolonged, low-irradiance

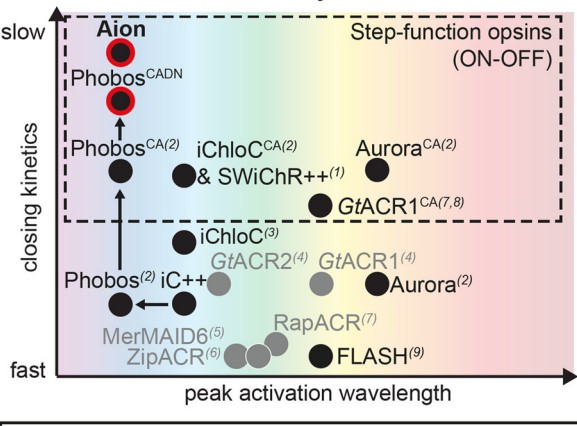

## Selection of currently available ACRs

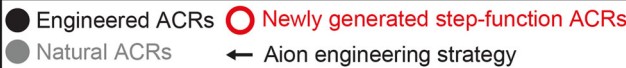

**Fig. 7 Aion broadens the available toolkit of optogenetic silencers in the temporal domain.** Schematic diagram showing a selection of the currently available ACRs used as optogenetic silencers in neurons. The ACRs are sorted according to their approximate peak activation wavelength (x-axis) and closing kinetics (y-axis). Engineered and natural ACRs are depicted as black and gray circles, respectively. Phobos[CADN] and Aion, the slow-cycling ACRs developed in this study, are highlighted with a red outline and the engineering strategy is indicated by black arrows. References to the published ACRs are indicated by numerical superscripts: 1[9], 2[11], 3[10], 4[13], 5[67], 6[72], 7[20], 8[73], 9[64].

illumination. In this way, unintended activation of fast cycling opsins, such as Chrimson, is avoided, while Aion accumulates in the conducting state over time. Thus, mutually exclusive activation of two opsins is possible using a slow-cycling blue light-sensitive opsin and a fast-cycling red light-sensitive opsin.

The long interstimulus intervals during Aion-mediated inhibition can in principle also simplify optical monitoring of neuronal activity during the silencing period. Since illumination of the opsin is required only transiently, imaging or photometry experiments could be performed in the dark periods, thus avoiding light artefacts. Certainly, red-sensitive indicators must be used to avoid inadvertent Aion activation. However, a more serious problem is to avoid accelerated channel closing with red-shifted excitation light of those indicators, given that inactivation of Aion is highly efficient already at 550 nm. Thus, the wide spectral separation of dark state and conducting state seems to be a disadvantage for the combination with commonly used reporter fluorophores, leaving only an imaging window compatible with far red-shifted sensors, such as the calcium indicator NIR-GECO[51] or the QuasAr voltage sensor family[52,53].

Aion showed efficient membrane localization and did not alter neuronal biophysical properties when expressed at moderate levels. However, in some cases local intracellular aggregations of the opsin were observed, similar to other first-generation opsins, such as NpHR[30] and GtACR2[29]. While expression levels and thus, overexpression artefacts can be well-controlled with single-cell electroporation, this is not trivial when using viral injections or transgenic lines. Thus, trafficking-optimization is required to avoid aggregation of Aion in intracellular organelles that could potentially lead to cell stress and toxicity. Insertion of the canonical sequences for ER export and membrane trafficking from Kir2.1 improved membrane expression of Aion, avoiding intracellular accumulations. Unlike previous reports[29,30], we did not observe larger photocurrents or stronger silencing in this trafficking-enhanced version of Aion. However, we observed

similar potency already at a fourfold lower DNA concentration, compared to the original Aion version. This indicates, that maximal membrane expression was reached in Aion, when aggregates were observed and that the same levels of membrane expression could be reached with less DNA and without aggregates using the trafficking-enhanced version.

While efficient membrane expression is desired in the somatodendritic compartment, axonal localization of ACRs should be avoided. In some axons and synaptic terminals, elevated Cl⁻ concentrations are reported[54–56], thus potentially leading to depolarization upon activation of ACRs. Indeed, local activation of GtACR2 at axon terminals led to presynaptic neurotransmitter release[6,57] and antidromic spiking[29,58]. Soma-targeting of GtACR2 avoided axonal expression and enhanced membrane expression in the somatodendritic compartment[29,57]. We therefore applied a similar strategy to generate the soma-targeted Aion variant somAion. C-terminal attachment of the soma-targeting signal from Kv2.1[31] led to restricted expression of somAion in the soma and proximal dendrites of CA1 pyramidal neurons. Moreover, as reported previously, this modification enhanced the silencing capacity of somAion compared to the original Aion or the trafficking-optimized Aion. This is likely due to the high-density expression of somAion in the somatodendritic compartment, efficiently shunting depolarization close to the axon initial segment and thus efficiently blocking AP generation.

Although soma-targeting of ACRs efficiently restricts their expression to the somatodendritic compartment, thereby overcoming the issue of axonal excitation and antidromic spiking, it limits the application of ACRs to somatic inhibition. If silencing AP initiation is the main goal, soma-targeted ACRs may be the tools of choice. However, if silencing of dendritic inputs anywhere in the cell is required, a non-soma-targeted, trafficking-optimized ACR is preferable. Moreover, ACR-mediated somatic inhibition does not allow local silencing of presynaptic terminals, which is required to investigate the role of specific projections from one neuronal population to its target region. In this case, alternative silencing methods should be considered. Inspired by potent inhibition via $G_{i/o}$ signaling of G-protein coupled receptors (GPCRs), such as $GABA_B$ receptors or the chemogenetic silencing tool hM4Di[59,60], $G_{i/o}$-coupled rhodopsin GPCRs were recently demonstrated as potent tools for presynaptic silencing. These include eOPN3, a mosquito-derived homolog of the mammalian encephalopsin[61] and a lamprey parapinopsin (PPO)[62]. Given the universal role of K⁺ ions for membrane hyperpolarization, a potent and selective K⁺ ChR with step-function properties may represent a solution to the limitations indicated above and become the tool of choice for silencing any neuronal compartment at any developmental stage. Until then, slow-cycling ACRs likely represent the major strategy for long-lasting somatodendritic silencing of neurons with millisecond precision.

We systematically compared the performance of Aion to GtACR1[CA], the latter being a slow-cycling variant engineered from the naturally-occurring GtACR1 by introduction of the point mutation C102A[19,20], which is homologous to the mutation C128A in Aion. However, in contrast to Aion, where the C128A mutation not only slows down channel closing kinetics, but also enables accelerated channel closure upon illumination with red-shifted light, accelerated channel closure was not possible with GtACR1[CA]. In fact, when red light was delivered at a time point when GtACR1[CA]-mediated photocurrents had almost decayed back to baseline (e.g. 198 s after initial opening), an increase in Cl⁻ conductance was observed, suggesting that red light brings GtACR1[CA] to an intermediate conducting state in the photocycle rather than back to the dark state. Hence, GtACR1[CA] exhibits slow-cycling but not truly step-function properties, hindering its applicability for neuronal inhibition with precisely timed on- and offset.

The second point mutation that confers Aion its long-lasting open state is on the residue D156 of the DC-gate. However, this residue is not conserved in natural ACRs, and most frequently replaced by a serine (S130)[19,63]. Mutation of the corresponding S130 to alanine in GtACR1 did not show a strong effect on the off-kinetics[19]. The different role of key residues in natural ACRs compared to CCR-derived engineered ACRs emphasizes the differences in photocycle and gating mechanisms. Certainly, structure-function insights derived from crystallography studies, such as the work by H. Kato, Y. Kim and colleagues who resolved the crystal structures of iC++ and GtACR1, are key to determine which properties and mutations are transferable between engineered and natural ACRs[63,64]. Using structure-guided approaches to combine the high single-channel conductance of the natural cryptophyte GtACRs with the step-function properties of CCR-derived ACRs hopefully leads the way to new, more potent step-function ACRs.

## Methods

**Molecular cloning of ACR variants.** For HEK-cell expression, mutations were introduced in previously described ACR constructs of GtACR1[65] (Addgene #85464) and Phobos$^{CA}$[11] (Addgene #98169) using the QuikChange II kit (Agilent Technologies, Santa Clara, CA). Successful single point mutagenesis was verified by conventional Sanger sequencing.

For neuronal expression, all engineered ACRs (see Table 1 for details) were cloned into an AAV2-backbone behind a human synapsin (hSyn) promoter, followed by the sequence for a citrine fluorescent protein. Phobos$^{CA}$ mutants (D156C/Aion, D156N, D156S, D156H) were generated by site-directed mutagenesis of our previously reported plasmid Phobos$^{CA}$-citrine[11] (Addgene #98218). The membrane-trafficking optimized variant of Aion, namely Aion-ts-citrine-ER was generated by adding a trafficking signal (ts: KSRITSEGEYIPLDQIDINV) from the Kir2.1 channel between the Aion and the citrine coding sequences, and the Kir2.1 ER export signal (FCYENV) following the C-terminus of citrine (pAAV-hSyn-Aion-ts-citrine-ER). The soma-targeted variant, somAion, was generated by fusing a trafficking signal from the potassium channel Kv2.1[31] to the C-terminus of Aion-citrine (pAAV-hSyn-Aion-citrine-Kv2.1).

**Patch-Clamp experiments in HEK-293 cells.** HEK-293 culture (ECACC 85120602, Sigma-Aldrich, Munich, Germany), transfection and electrophysiological recordings[66,67] were performed as follows: cells were seeded at a density of $10^5$ /ml on poly-D-lysine–coated coverslips, supplemented with 1 μM all-trans retinal, and were transiently transfected using Fugene HD (Promega, Madison, WI) one day post seeding. 24–48 days post-transfection, whole-cell patch-clamp recordings were performed at 22 °C using fire-polished patch pipettes (1.5–2.5 MΩ pipette resistance), and a 140 mM NaCl agar bridge reference electrode. Patch pipettes were filled with (in mM): 2 MgCl$_2$, 2 CaCl$_2$, 1 KCl, 1 CsCl, 10 EGTA, 10 HEPES, 110 Na-Aspartate, while the bath solution contained: 2 MgCl$_2$, 2 CaCl$_2$, 1 KCl, 1 CsCl, 10 HEPES, 140 NaCl, both pH 7.2 adjusted with N-methyl-D-glucamine or citric acid. The buffer osmolarity was set with glucose to 320 mOsm (extracellular) or 290 mOsm (intracellular). Cells were measured at 0 mV membrane potential, corrected for a liquid junction potential of −10.5 mV. Only cells with an access resistance below 10 MΩ and at least 30-fold higher membrane resistance where measured. Signals were amplified (AxoPatch200B), digitized (DigiData1440A), and acquired using Clampex 10.4 (Molecular Devices, Sunnyvale, CA). Light from a Polychrome V (TILL Photonics, Planegg, Germany) with 7 nm bandwidth guided into an Axiovert 100 microscope (Carl Zeiss, Jena, Germany) was used for optogenetic stimulation. For action spectra recordings a 150 W Xenon lamp (LOT-QuantumDesign, Darmstad, Germany), bandpass filtered to 600 ± 10 nm was coupled into the light path using a 30/70 beam splitter (Chroma, Bellows Falls, VT) to accelerate channel closing. Light application was controlled via a programmable shutter system (VS25 and VCM-D1; Vincent Associates, Rochester, NY) for both light sources. Irradiances were measured in the sample plane using a calibrated photodiode power sensor (S130VC, Thorlabs, Bergkirchen, Germany) after passing through a W Plan-Apochromat 40×/1.0 DIC objective (0.066 mm$^2$ illuminated field, Carl Zeiss).

**Preparation of organotypic hippocampal slice cultures and transgene delivery.** All procedures were in agreement with the German national animal care guidelines and approved by the independent Hamburg state authority for animal welfare (Behörde für Justiz und Verbraucherschutz). They were performed in accordance with the guidelines of the German Animal Protection Law and the animal welfare officer of the University Medical Center Hamburg-Eppendorf.

Organotypic hippocampal slices were prepared from Wistar rats at post-natal day 5–7[68] as follows: dissected hippocampi were cut into 350 μm slices with a tissue chopper (McIlwain type 10180, Ted Pella) and placed on a porous membrane

**Table 1 List of DNA constructs used for experiments in organotypic hippocampal slices.**

| DNA construct | Concentration (ng/μl) |
|---|---|
| hSyn-Phobos$^{CADC}$-citrine (Aion) | 20 |
| hSyn-Aion-citrine-Kv2.1 (somAion) | 20 |
| hSyn-Aion-ts-citrine-ER | 5 |
| hSyn-Phobos$^{CADN}$-citrine | 20 |
| hSyn-Phobos$^{CADS}$-citrine | 20 |
| hSyn-Phobos$^{CADH}$-citrine | 20 |
| hSyn-GtACR1$^{CA}$-citrine | 5 |
| hSyn-mCerulean | 50 |
| hSyn-citrine | 20 |

Plasmids were used for single-cell electroporation of CA1 neurons at the indicated concentrations.

(Millicell CM, Millipore). Cultures were maintained at 37 °C, 5% CO$_2$ in a medium containing 80% MEM (Sigma M7278), 20% heat-inactivated horse serum (Sigma H1138) supplemented with 1 mM L-glutamine, 0.00125% ascorbic acid, 0.01 mg/ml insulin, 1.44 mM CaCl$_2$, 2 mM MgSO$_4$ and 13 mM D-glucose. Slice cultures were allowed to mature for at least 2 weeks in the incubator before experimental assessment. No antibiotics were added to the culture medium. Pre-warmed medium was replaced twice per week.

For transgene delivery in organotypic slices, individual CA1 pyramidal cells were transfected by single-cell electroporation between DIV 14–16[69]. With exception for pAAV-hSyn-GtACR1-C102A and pAAV-hSyn-Aion-ts-citrine-ER, which were used at a final concentration of 5 ng/μl, all plasmids (see Table 1 for details) were used at a final concentration of 20 ng/μl in K-gluconate-based solution consisting of (in mM): 135 K-gluconate, 10 HEPES, 4 Na$_2$-ATP, 0.4 Na-GTP, 4 MgCl$_2$, 3 ascorbate, 10 Na$_2$- phosphocreatine (pH 7.2). A plasmid encoding pCI-hSyn-mCerulean (at 50 ng/μl) was co-electroporated with the opsin plasmids and served as a morphology marker. Once the pipette containing the plasmid mix was touching the membrane of the target cell, an Axoporator 800 A (Molecular Devices) was used to deliver 50 hyperpolarizing pulses (−12 V, 0.5 ms) at 50 Hz. During electroporation slices were maintained in pre-warmed (37 °C) HEPES-buffered solution (in mM): 145 NaCl, 10 HEPES, 25 D-glucose, 2.5 KCl, 1 MgCl$_2$ and 2 CaCl$_2$ (pH 7.4, sterile filtered).

**Two-photon imaging of neurons in slice cultures.** Neurons in organotypic slice cultures (DIV 19-21) were imaged with two-photon microscopy to check for the live expression and subcellular localization of the different ACRs. The custom-built two-photon imaging setup was based on an Olympus BX-51WI upright microscope upgraded with a multiphoton imaging package (DF-Scope, Sutter Instrument), and controlled by ScanImage 2017b software (Vidrio Technologies). Fluorescence was detected through the objectives (Leica HC FLUOTAR L 25x/0.95 W VISIR or LUMPLFLN 60XW, Olympus) and the oil immersion condenser (NA 1.4) by two pairs of GaAsP photomultiplier tubes (Hamamatsu, H11706-40). Dichroic mirrors (560 DXCR, Chroma Technology) and emission filters (ET525/70m-2P, ET605/70m-2P, Chroma Technology) were used to separate green and red fluorescence. Excitation light was blocked by short-pass filters (ET700SP-2P, Chroma Technology). A tunable Ti:Sapphire laser (Chameleon Vision-S, Coherent) was set to 810 nm to excite mCerulean. An Ytterbium-doped 1070-nm pulsed fiber laser (Fidelity-2, Coherent) was used at 1070 nm to excite citrine, which was detected in the green channel. The open-source software Fiji[70] was used for visualization and processing of images.

**Electrophysiology of neurons in slice cultures.** At DIV 19-21, whole-cell patch-clamp recordings of opsin-transfected or wt CA1 pyramidal neurons were performed. Experiments were done at room temperature (21–23 °C) under visual guidance using a BX 51WI microscope (Olympus) equipped with Dodt-gradient contrast and a Double IPA integrated patch amplifier controlled with SutterPatch software (Sutter Instrument). Patch pipettes with a tip resistance of 3-4 MΩ were filled with intracellular solution consisting of (in mM): 135 K-gluconate, 4 MgCl$_2$, 4 Na$_2$-ATP, 0.4 Na-GTP, 10 Na$_2$-phosphocreatine, 3 ascorbate, 0.2 EGTA, and 10 HEPES (pH 7.2). Artificial cerebrospinal fluid (ACSF) was used as extracellular solution in the recording chamber and consisted of (in mM): 135 NaCl, 2.5 KCl, 2 CaCl$_2$, 1 MgCl$_2$, 10 Na-HEPES, 12.5 D-glucose, 1.25 NaH$_2$PO$_4$ (pH 7.4). In all experiments synaptic transmission was blocked by adding 10 μM CPPene, 10 μM NBQX, and 100 μM picrotoxin (Tocris) to the recording solution. Measurements were corrected for a liquid junction potential of −14.5 mV. Access resistance of the recorded neurons was continuously monitored and recordings above 30 MΩ were discarded. A 16 channel LED light engine (CoolLED pE-4000) was used for epi-fluorescence excitation and delivery of light pulses for optogenetic stimulation (ranging from 385 to 635 nm). Irradiance was measured in the object plane with a 1918 R power meter equipped with a calibrated 818 ST2 UV/D detector (Newport,

Irvine CA) and divided by the illuminated field of the Olympus LUMPLFLN 60XW objective (0.134 mm$^2$).

For photocurrent density measurements in voltage-clamp mode CA1 cells expressing Phobos$^{CADN}$, Aion or GtACR1$^{CA}$ were held at −35 mV to detect outward Cl$^−$ currents elicited by blue (460 nm, 20 ms, 10 mW/mm$^2$) or green light (525 nm, 20 ms, 10 mW/mm$^2$), and terminated by orange (595 nm, 1 s, 10 mW/ mm$^2$) or red light (660 nm, 1 s, 10 mW/mm$^2$). To calculate photocurrent densities, the peak and stationary (at 30 and 120 s after light offset) photocurrent amplitudes (in pA) were divided by the cell membrane capacitance (in pF) which was automatically recorded by the SutterPatch software in voltage-clamp mode ($V_{hold} = −75$ mV).

In current-clamp experiments holding current was injected to maintain CA1 cells near their resting membrane potential (−75 − −80 mV). To assess the capability of Phobos$^{CADN}$, Aion and GtACR1$^{CA}$ to block AP firing, a light pulse at their respective peak activation wavelength (460 or 525 nm, 20 ms, 10 mW/mm$^2$) was delivered after 5 s during a train of somatic current step injections (300 pA, 2 s, 2.5 s ISI, for 55 s). To assess accelerated termination of silencing for each ACR, a light pulse at their respective peak inactivation wavelength (595 or 660 nm, 1 s, 10 mW/mm$^2$) was delivered after 40 s. Spiking rate was calculated before opening (0–5 s), after opening (5–40 s) and after closing (45–50 s) of the ACR by dividing the number of current-evoked APs by the respective time interval.

To measure the ability of Phobos$^{CADN}$, Aion, Aion-ts-citrine-ER and somAion to shift the rheobase upon light-activation, 13 depolarizing current ramps (2 s, from 0–600 pA up to 0–1000 pA, 2.5 s ISI) were injected into CA1 neurons over 1 min in dark and light conditions (1 s light pulse of 460 nm between 1st and 2nd current ramp injections) at irradiance values ranging from 0.001 to 10 mW/mm$^2$. For each ramp, the injected current at the time of the first AP was defined as the rheobase. The rheobase shift over time was calculated by subtracting the rheobase of each ramp after light stimulation from the rheobase value before light stimulation (1st ramp). The relative change in the number of ramp-evoked APs was calculated counting the total number of APs elicited during each current ramp injection after light stimulation and normalized to the number of APs elicited at the same time point in the absence of light. The same experiment was conducted for GtACR1$^{CA}$, but using 525 nm light.

For Aion and GtACR1$^{CA}$ the ability to block AP firing was further quantified over a period of 5 min. For this, the change in membrane depolarization (after median filtering of the raw voltage traces) during somatic current step injections was measured before and after a brief light stimulus (1 s, 10 mW/mm$^2$, Aion: 460 nm, GtACR1$^{CA}$: 525 nm). In long-term silencing experiments inside the incubator, 35 mm diameter petri dishes containing a Millipore membrane with two organotypic slices on top of 1 ml of cell culture medium were placed on a custom-made LED chamber controlled by an Arduino board. After overnight (12 h) blue-light stimulation (3 s 460-nm light pulses every 5 min, 0.3 mW/mm$^2$), slices were transferred to the patch clamp setup and whole-cell current-clamp recordings were performed in Aion-expressing cells to evaluate baseline spiking performance and functionality of Aion-mediated silencing. Residual Aion conductance was terminated by applying orange light before the start of the recordings.

Passive and active membrane parameters were measured after O/N blue light stimulation in Aion-citrine-, citrine-only-expressing and non-transduced wt CA1 pyramidal cells. As an additional control, the same measurements were done in Aion-citrine-expressing cells which were not stimulated with light. Resting membrane potential, membrane resistance and capacitance were automatically recorded by the SutterPatch software in voltage-clamp mode ($V_{hold} = −75$ mV) in response to a voltage test pulse of 100 ms and −5 mV. The number of elicited APs were counted in response to a somatic current injection of 300 pA in current-clamp mode (0 pA holding current). For the 1st elicited AP, the voltage threshold, peak and amplitude were measured.

**Drosophila maintenance and generation of transgenes.** All fly stocks were maintained at 25 °C and 70% rel. humidity on standard cornmeal/molasses food. All Drosophila stocks were obtained from the Bloomington Drosophila Stock Center unless indicated otherwise (BDSC, Bloomington, Indiana, USA). The following lines were used: A08n-splitGal4, 82E12-Gal4 (2nd chr.), UAS-Kir2.1[33], 27H06-LexA (BDSC #54751), LexAop-CsChrimson (BDSC #55139), UAS-GtACR1 (BDSC #92983). Aion constructs (Aion-smGFP, Aion-ts-mScarlet-ER, somAion-citrine) were cloned into Drosophila transgenic vectors (pUAST-AttB with 5xUAS, pJFRC7 with 20xUAS) and inserted into the attP2 landing site by phiC31-mediated trangenesis[71] (BestGene, USA).

**Optogenetic behavioral assays in Drosophila.** Larvae expressing CsChrimson in C4da and Aion in A08n neurons (82E12-Gal4, 27H06-LexA/+; UAS-CsChrimson/ UAS-Aion) were grown in darkness on grape agar plates with yeast paste containing 5 mM all-trans-retinal (Sigma-Aldrich). Staged third instar larvae (96 h ± 3 h AEL) were carefully transferred under low red-light conditions to a 2% agar plates with a 1 ml water film. CsChrimson was activated with 635 nm light (760 µW/cm$^2$) for 5 s. Aion was activated with 460 nm and intensities of 0.08- 8 mW/cm$^2$ for up to 30 s and inactivated with 595 nm at 7 mW/cm$^2$. Between each activation steps, 30 s of no light were included, to allow full functional recovery of CsChrimson. Videos were taken during the experiment and analyzed using the Fiji cell counter plugin (ImageJ, NIH, Bethesda). Staging, experiments, and analyses

were done in a blinded fashion. Rolling was defined as at least one complete 360° roll. Bending was defined as a c-shape like twitching, typically seen before rolling behavior. Turning behavior describes head turning and thereby a direction changes of locomotion. Stop behavior describes a stopping of movement, no change in behavior means no change in behavior. Stop and turn, stop and no change in behavior were defined as non-nociceptive behavior. All behavioral assays and analyses were performed in a blinded and randomized fashion.

**Mechanonociception assays in Drosophila.** Mechanonociception experiments were performed with calibrated von-Frey-filaments (50 mN) and staged third instar larvae (96 h AEL ± 3 h) expressing Aion, GtACR1 or Kir2.1 in A08n neurons (A08n-splitGal4/UAS-Aion or UAS-GtACR1 or UAS-Kir2.1). Larvae were carefully transferred to a 2% agar plates with a 1 ml water film and stimulated twice on mid-abdominal segments (a3–a6) within 2 s. Behavioral responses (non-nociceptive (no response, stop, stop and turn), bending, rolling) were noted and only behavioral responses to the second stimulus were analyzed and plotted. Staging and experiments were done in a blinded fashion and randomized fashion.

**Drosophila immunohistochemistry and confocal microscopy.** Third instar larvae were dissected in dissection buffer (108 mM NaCl, 5 mM KCl, 4 mM NaHCO$_3$, 1 mM NaH$_2$PO$_4$, 5 mM Trehalose, 10 mM Sucrose, 5 mM HEPES, 8.2 mM MgCl$_2$, 2 mM CaCl$_2$, pH 7.4), fixed with 4% formaldehyde/PBS for 15 min, washed with PBS + 0.3 Triton-X, antibody stained, and mounted on poly-l-lysine (Sigma)-coated cover slips in SlowFade Gold (Thermo Fisher, Carlsbad, CA, USA). Aion-smGFP-HA expression in A08n neurons was visualized by rat anti-HA (3F10, 1:100, Sigma-Aldrich) together with mouse anti-Fas3 antibody (7G10, 1:100, DSHB, IA, USA) immunostaining to visualize sensory axons. Appropriate secondary donkey antibodies conjugated to DyLight488 and Cy3, respectively, were used at 1:300 dilution (Jackson ImmunoResearch, Cambridgeshire, UK). Larval brains were imaged by confocal microscopy (Zeiss LSM700, Zeiss, Jena) and processed in Fiji (ImageJ, NIH, Bethesda[70]).

**Statistics and Reproducibility.** Statistical analyses were performed using GraphPad Prism 9.0. Not normally distributed data were tested for significant differences (*$p < 0.05$, **$p < 0.01$, ***$p < 0.001$) using either the nonparametric, two-sided Mann–Whitney test (two groups, non-matching values), the Friedman test (more than two groups, matching-values), or the Kruskal–Wallis test (more than two groups, non-matching values) followed by Dunn's multiple comparisons test, as indicated in the figure legends. For HEK cell and neuronal recordings individual datapoints are shown together with the mean ± standard error of the mean (SEM), or median, as indicated. Given n numbers represent biological replicates (i.e. HEK-cells, neurons, Drosophila larvae). For behavioral analysis of Drosophila larvae a chi-square test was performed. Source data and details on the statistical tests are provided in Supplementary Data 1.

**Reporting summary.** Further information on research design is available in the Nature Research Reporting Summary linked to this article.

## Data availability

Source data are provided with this paper. All data generated in this study are provided in Supplementary Data 1.

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

## Acknowledgements
We thank Meike Petersen, Maila Reh and Tharsana Tharmalingam for excellent technical support. Oren Princz-Lebel and Alberto J. González Hernández helped with initial characterization of Phobos[CADC] at the 2019 Cajal Course "Biosensors and Actuators in Cellular and Systems Neuroscience" and suggested the name "Aion". This work was funded by the German Research Foundation (SPP 1926, Project Nr.: 315380903 WI4485/3-2 to J.S.W. and SO 1337/2-2 to P.S., SFB 1078 B2 to P.H.). P.H. is a Hertie Senior Professor for Neuroscience supported by the Hertie Foundation.

## Author contributions
Conceptualization: S.R.R., J.W., P.S., J.S.W.; Data Acquisition: S.R.R., J.W., F.T., N.D.; Analysis: S.R.R., J.W., F.T.; Methodology: S.R.R., J.W., F.T., K.S., N.D., P.S., J.S.W.; Supervision: P.H., P.S., J.S.W.; Funding Acquisition: P.H., P.S., J.S.W.; Project Administration: J.S.W.; Writing: S.R.R., J.S.W. with help from all other authors.

## Funding

## Competing interests
The authors declare no competing interests.
