## [Peer Review File · Communications Biology]

Reviewers' comments:

Reviewer #1 (Remarks to the Author):

Rodrigues-Rozada and colleagues describe directed point mutation experiments designed to produce switchable anion-conducting channelrhodopsins with long-lasting photoinhibition. Three derived channels were then extensively characterised *in vitro* and *in vivo*. Further enhancements of membrane trafficking and somal targeting were applied and characterised in the most effective eACR, Aion. The rationale for the experiments is clear and meets an important need in experimental neuroscience. Consequently, the work has broad application. The characterization is comprehensive and provides strong validation of the novel channels. The data are presented extremely well, and the authors are to be congratulated for their beautiful images and high-quality work.

In relation to Phobos and Aion, the channels were initially created by directed mutation of the ion channel to alter selectivity from cation to anion. This manuscript describes further changes to affect channel opening kinetics. Clearly even after these new alterations, the photoinhibition is obvious. However, I would be interested in the authors consideration relating to ion selectivity and whether there is any chance the new alterations could have altered that. I note this was tested for the original Phobos in the 2017 Scientific Reports paper (Wietek et al.,). Whilst the mutations are carefully chosen to be in the DC gate, the extent of conformational change in the structure of a complex protein is not always predictable. None of the measurements in the current paper address the ion selectivity issue. The recording conditions, for example in the slice preparations where neurons are held at -35mV, strongly favor Cl⁻ entry (equilibrium potential in low -90's) but would also favor K⁺ exit (equilibrium potential in low -100's). In these conditions, any Na⁺ entry would just reduce the size of the observed outward current. As stated above, the effect is clearly inhibitory and so one might say why worry about ion selectivity. However, there are many examples where *in vivo*, the ion gradients are different and maintenance of those gradients compromised in different neuronal compartments, such that the overall effect, and subsequent result interpretation, might be altered depending upon the ion selectivity. It seems to me that at the initial characterization of a tool that should have widespread application, it might be worth demonstrating the ion channel selectivity, particularly as the experimental proof just requires assessment of photocurrents under different recording conditions. It is not my view that this falls into a category of major concern – I am sufficiently convinced by the examples presented – but I provide it as a likely concern and would be interested in the authors views.

Minor Issues

The first sentence of the Abstract requires attention.

Page 4, line 127 mentions a trace for GtACR1CA in Figure S2C. I don't see this trace in the Figure.

Page 8, line 221. The lack of effect of red light in restoring AP following GtACR1 photoinhibition (figure 3B) is congruent with the data from HEK cells. It is not clear to me why, over that timeframe, the reduction in current through GtACR1 is not sufficient to observe return of current injection induced APs?

Why is the expression in the A08n neurons of the *Drosophila* not shown?

The data in Figure 6 look convincing, and the number of observations provides a view that the data are robust. However, there is no evidence provided in relation to the variance. Without experience in the method, it is hard to understand whether these is a difference within the bending response, for example. Some explanation and description of how the data can be subjectively interpreted, in the absence of a statistical measure, would be helpful. There is mention in the methods of a chi-square test being applied, but it is not clear where the results are described.

Reviewer #2 (Remarks to the Author):

The ms describes an optogenetic tool to silence neurons. The tool is based on the blue shifted step function anion-conducting ChR variant PhobosCA and is called Aion. Aion is bistable and can be switched on and off with blue and orange light within milliseconds. It reveals a long open state and slow spontaneous closing time, which allows for long-term silencing of neuronal networks demonstrated in organotypic hippocampal cultures and nociception in larvae of *Drosophila melanogaster*.

Experimental results are presented in 6 Figures.

1. Characterization of different mutations at D156 of Phobos reveal that D156C mutation (Aion) reveals slower closing time in comparison to control constructs
2. Expression of Aion in pyramidal neurons and blue light activation induces a sustained photocurrent, which can be switched off by orange light. The light-induced photocurrent does not decline to baseline levels in comparison to control constructs.
3. Aion reduces efficiently AP firing for a longer time period in comparison to GtACR1. No differences are observed between Aion and GtACR1 in the light intensity necessary for sufficient silencing.
4. Aion silences CA1 pyramidal neuron firing over long-time period (>10min) using short blue light pulses every 5 min.
5. For somatodendritic targeting Aion was tagged with a fluorescent protein (Citrine) and Kv2.1 targeting sequences.
6. Expression of Aion in larvae from *Drosophila* inhibits nociceptive circuit function

The experiments are technically sound and the results are nicely presented. Aion extends the family of silencer tools. Due to the slow spontaneous closing time achieved via D156C mutation Aion can be used to silence neuronal networks over long time period with minimal light stimulation. I have no major comments.

I recommend this manuscript for publication in *Communications Biology*.

Reviewer #3 (Remarks to the Author):

Anion channelrhodopsins (ACR) are powerful tools for the inhibitory optogenetics to efficiently inhibit neural spiking by low-power light. While various ACRs and their site-directed mutants are known to exhibit a large photocurrent, red-shifted absorption, fast off kinetics, and so on, no ideal SFO-type ACR was known to close at a different wavelength than the activation light and to have a sufficiently long opening time, despite its usefulness for long-period optical neural control.

In this study, Dr. Rodriguez-Rozada succeeded in developing a new SFO-type ACR (Aion) by introducing a site-directed mutation in an asparagine residue (D156) that constitutes the DC gate, which is known to be related to the photocyclic turnover rate. After photoactivation, Aion keeps its open state for more than 15 min without subsequent photons. The authors carefully compared the properties of Aion with other SFO-type ACRs in the cultured neurons, and found that Aion is able to stably inhibit neural activity for a much longer period of time than the others. Also, they showed that Aion nicely works in *Drosophila* larvae with low-power light and enables us all-optical control in a combination with CsChrimson.

The experimental methods and statistical analyses were carefully designed and the interpretation based on the results is solid so that this work potentially publishable in Commun. Biol. However, I would like to suggest a few points which need to be addressed by the authors.

-Major points

Figure 2

Why do the photocurrents of Aion and PhobosCADN overshoot to lower than the baseline just after the illumination of orange light? In addition, why does the former become slightly positive (a residual channel activity remains even after the orange-light illumination)?

Page 11, Line 292-294

"Whole cell current-clamp recordings in Aion-expressing cells after 12 h of light stimulation showed that depolarization-induced APs (2 s current injections every 4.5 s)..."

Since action potentials were observed during the first current injection, I think that the neuron is recovered from the inhibition by opened Aion during 12-h light stimulation. However, it should need a long time to wait the opened Aion closes before the electrophysiological assay. It would be useful to describe how long time the authors waited between the 12-h light stimulation and the assay here or in the Materials and Methods.

Figure 4 I

Although the membrane resistance was indicated as not significant between four cell types, the mean and SEM looks to be substantially different between the O/N light stim cells. and the Not transfected/Citrine only cells. How much the p-value between them? If the difference is significant, it needs to be explained in the main text.

-Minor points

Page 4, Line 125

"...the 500 ms illumination (Fig. 1B)..."

The 500 ms illumination would not be in Fig. 1B, but in Fig. S2B.

Page 4, Line 128

"Fig. S2C" would be a typo of "Fig. S2B"?

Figure 6E

"Chrimson" should be written as "CsChrimson" to avoid misunderstanding.

Reviewer #1 (Remarks to the Author):

Rodrigues-Rozada and colleagues describe directed point mutation experiments designed to produce switchable anion-conducting channelrhodopsins with long-lasting photoinhibition. Three derived channels were then extensively characterised in vitro and in vivo. Further enhancements of membrane trafficking and somal targeting were applied and characterised in the most effective eACR, Aion. The rationale for the experiments is clear and meets an important need in experimental neuroscience. Consequently, the work has broad application. The characterization is comprehensive and provides strong validation of the novel channels. The data are presented extremely well, and the authors are to be congratulated for their beautiful images and high-quality work.

We thank the reviewer for acknowledging the quality of our work.

In relation to Phobos and Aion, the channels were initially created by directed mutation of the ion channel to alter selectivity from cation to anion. This manuscript describes further changes to affect channel opening kinetics. Clearly even after these new alterations, the photoinhibition is obvious. However, I would be interested in the authors consideration relating to ion selectivity and whether there is any chance the new alterations could have altered that. I note this was tested for the original Phobos in the 2017 Scientific Reports paper (Wietek et al.,). Whilst the mutations are carefully chosen to be in the DC gate, the extent of conformational change in the structure of a complex protein is not always predictable. None of the measurements in the current paper address the ion selectivity issue. The recording conditions, for example in the slice preparations where neurons are held at -35mV, strongly favor Cl⁻ entry (equilibrium potential in low -90's) but would also favor K⁺ exit (equilibrium potential in low -100's). In these conditions, any Na⁺ entry would just reduce the size of the observed outward current. As stated above, the effect is clearly inhibitory and so one might say why worry about ion selectivity. However, there are many examples where in vivo, the ion gradients are different and maintenance of those gradients compromised in different neuronal compartments, such that the overall effect, and subsequent result interpretation, might be altered depending upon the ion selectivity. It seems to me that at the initial characterization of a tool that should have widespread application, it might be worth demonstrating the ion channel selectivity, particularly as the experimental proof just requires assessment of photocurrents under different recording conditions. It is not my view that this falls into a category of major concern – I am sufficiently convinced by the examples presented – but I provide it as a likely concern and would be interested in the authors views.

We agree with the reviewer. It is important to show that mutations in the DC gate did not alter ion selectivity of Aion compared to its parental opsin Phobos^{CA}. Indeed, mutations of Aspartate 156 and its homologs in cation conducting ChRs can impose a shift on the ratio of conducted cations. Moreover, since Aspartate 156 in ChR2 is located close to the protonated retinal Schiff base, its mutation may also affect the spectral properties of the opsin by changing the energy landscape in the retinal binding pocket. We therefore added a basic characterization of the current-voltage relationship of Aion photocurrents as well as a characterization of activation and inactivation spectra in HEK293 cells. Reversal potential for Aion was similar to Phobos^{CA} and matched the Nernst potential of chloride in our experimental conditions. Yet, we found slight alterations of the action spectra of Aion, likely due to the changed retinal binding pocket, as pointed out above.

The new data are shown in a new supplemental figure (Fig. S3, see also below) and mentioned in the main text (lines 163-170).

New Fig. S3

Minor Issues

1. The first sentence of the Abstract requires attention.

We changed the first sentences of the abstract (lines 24-26).

2. Page 4, line 127 mentions a trace for GtACR1CA in Figure S2C. I don't see this trace in the Figure.

We accidentally referred to the wrong figure panel. This is now corrected. Please note that we also fixed a second referencing error in this paragraph (lines 130-134).

3. Page 8, line 221. The lack of effect of red light in restoring AP following GtACR1 photoinhibition (figure 3B) is congruent with the data from HEK cells. It is not clear to me why, over that timeframe, the reduction in current through GtACR1 is not sufficient to observe return of current injection induced APs?

Over the time course of the recording shown in figure 3B (50 s after channel opening), GtACR1^{CA} photocurrents do not fully revert to zero (see also figs. 1D,E and 2B,D). The residual photocurrent was sufficient to still block the depolarization-evoked spikes, because the currents injected via the patch pipette were driving the membrane voltage just above spike threshold in baseline conditions. The point of this figure was to show that (residual) photocurrents of GtACR1^{CA} could not be terminated by red light. To better quantify the decline of the silencing capacity over time, we therefore did the rheobase experiments (Fig. 3C-J). This experiment confirms the observation in 3B, namely that, while decreasing over time, the effect on APs is not yet completely gone 50s after light.

We changed the text in the manuscript to make this point clearer (lines 244-245).

4. Why is the expression in the A08n neurons of the Drosophila not shown?

We apologize for the oversight. We added a confocal overview image of the Aion-expressing A08n neuron pair in panel A of figure 6 (see below).

Updated fig. 6

5. The data in Figure 6 look convincing, and the number of observations provides a view that the data are robust. However, there is no evidence provided in relation to the variance. Without experience in the method, it is hard to understand whether there is a difference within the bending response, for example. Some explanation and description of how the data can be subjectively interpreted, in the absence of a statistical measure, would be helpful. There is mention in the methods of a chi-square test being applied, but it is not clear where the results are described.

All data underlying the figures and all statistical analyses are now provided in a complementary file along with the manuscript. We did not include these data with the initial submission. The reader is now referred to these data sheets in the 'Statistics and Reproducibility' section. Moreover, we added asterisks to the plots indicating the conditions that differed significantly from each other.

Reviewer #2 (Remarks to the Author):

The ms describes an optogenetic tool to silence neurons. The tool is based on the blue shifted step function anion-conducting ChR variant PhobosCA and is called Aion. Aion is bistable and can be switched on and off with blue and orange light within milliseconds. It reveals a long open state and slow spontaneous closing time, which allows for long-term silencing of neuronal networks demonstrated in organotypic hippocampal cultures and nociception in larvae of *Drosophila melanogaster*.

Experimental results are presented in 6 Figures.

1. Characterization of different mutations at D156 of Phobos reveal that D156C mutation (Aion) reveals slower closing time in comparison to control constructs
2. Expression of Aion in pyramidal neurons and blue light activation induces a sustained photocurrent, which can be switched off by orange light. The light-induced photocurrent does not decline to baseline levels in comparison to control constructs.
3. Aion reduces efficiently AP firing for a longer time period in comparison to GtACR1. No differences are observed between Aion and GtACR1 in the light intensity necessary for sufficient silencing.
4. Aion silences CA1 pyramidal neuron firing over long-time period (>10min) using short blue light pulses every 5 min.
5. For somatodendritic targeting Aion was tagged with a fluorescent protein (Citrine) and Kv2.1 targeting sequences.
6. Expression of Aion in larvae from *Drosophila* inhibits nociceptive circuit function

The experiments are technically sound and the results are nicely presented. Aion extends the family of silencer tools. Due to the slow spontaneous closing time achieved via D156C mutation Aion can be used to silence neuronal networks over long time period with minimal light stimulation. I have no major comments.

I recommend this manuscript for publication in *Communications Biology*.

We thank the reviewer for the positive feedback.

Reviewer #3 (Remarks to the Author):

Anion channelrhodopsins (ACR) are powerful tools for the inhibitory optogenetics to efficiently inhibit neural spiking by low-power light. While various ACRs and their site-directed mutants are known to exhibit a large photocurrent, red-shifted absorption, fast off kinetics, and so on, no ideal SFO-type ACR was known to close at a different wavelength than the activation light and to have a sufficiently long opening time, despite its usefulness for long-period optical neural control.

In this study, Dr. Rodriguez-Rozada succeeded in developing a new SFO-type ACR (Aion) by introducing a site-directed mutation in an asparagine residue (D156) that constitutes the DC gate, which is known to be related to the photocyclic turnover rate. After photoactivation, Aion keeps its open state for more than 15 min without subsequent photons. The authors carefully compared the properties of Aion with other SFO-type ACRs in the cultured neurons, and found that Aion is able to stably inhibit neural activity for a much longer period of time than the others. Also, they showed that Aion nicely works in *Drosophila* larvae with low-power light and enables us all-optical control in a combination with CsChrimson.

The experimental methods and statistical analyses were carefully designed and the interpretation based on the results is solid so that this work potentially publishable in *Commun. Biol.* However, I would like to suggest a few points which need to be addressed by the authors.

We thank the reviewer for the careful evaluation of our manuscript.

-Major points

1. Figure 2

Why do the photocurrents of Aion and PhobosCADN overshoot to lower than the baseline just after the illumination of orange light? In addition, why does the former become slightly positive (a residual channel activity remains even after the orange-light illumination)?

The reviewer raises an important point. We did these experiments at a depolarized membrane potential (-35 mV) to obtain well-pronounced chloride currents. However, this configuration is quite artificial, as neurons would naturally not sit at such depolarized potentials for prolonged time. Maintaining a chloride conductance under such conditions may result in changes of chloride homeostasis and ionic balance in the cytoplasm, thus leading to slow drifts of membrane voltage (and consequently, holding current). Moreover, cytoplasmic accumulation of chloride likely activates chloride extrusion mechanisms, which become apparent as slow drifts of membrane voltage after channel closure. Under more natural conditions, when the neuron is sitting close to the resting membrane voltage, these issues are less prevalent. Therefore, we continued our characterization of ACR-mediated silencing with current clamp recordings in the remainder of the manuscript.

We now raise awareness to this aspect in the manuscript (lines 229-236).

2. Page 11, Line 292-294

“Whole cell current-clamp recordings in Aion-expressing cells after 12 h of light stimulation showed that depolarization-induced APs (2 s current injections every 4.5 s)...”

Since action potentials were observed during the first current injection, I think that the neuron is recovered from the inhibition by opened Aion during 12-h light stimulation. However, it should need a long time to wait the opened Aion closes before the electrophysiological assay. It would be useful to describe how long time the authors waited between the 12-h light stimulation and the assay here or in the Materials and Methods.

Thanks for pointing this out. Indeed, we illuminated Aion expressing neurons with orange light before the recordings to ensure that Aion was fully closed. This is now better explained in the main text (lines 313-320) and in the methods section (lines 797-801).

3. Figure 4 I

Although the membrane resistance was indicated as not significant between four cell types, the mean and SEM looks to be substantially different between the O/N light stim cells. and the Not transfected/Citrine only cells. How much the p-value between them? If the difference is significant, it needs to be explained in the main text.

There is heterogeneity in membrane resistance between the groups, but no group was significantly different vs any other. P values between the O/N stim Aion cells vs. non-transfected cells and O/N stim Aion cells vs. Citrine cells was 0.22 and 0.32, respectively. More importantly, there was no difference between Aion-expressing cells that were not stimulated vs. those stimulated O/N ($P > 0.99$), indicating that this chronic treatment did not alter membrane properties. We now mention the trend towards lower membrane resistance in Aion expressing cells and the absence of an O/N-light-mediated effect in the text (lines 327-329). All source data and statistical tests are provided now in a source data file along with the manuscript.

-Minor points

1. Page 4, Line 125

“...the 500 ms illumination (Fig. 1B)...”

The 500 ms illumination would not be in Fig. 1B, but in Fig. S2B.

Corrected

2. Page 4, Line 128

“Fig. S2C” would be a typo of “Fig. S2B”?

Corrected

3. Figure 6E

“Chrimson” should be written as “CsChrimson” to avoid misunderstanding.

Corrected

REVIEWERS' COMMENTS:

Reviewer #1 (Remarks to the Author):

The respectful response from the authors to all the reviewer's questions and concerns is greatly appreciated. The inclusion of the small pieces of additional data addresses all my concerns and strengthens the conclusions of the work. This is a very high quality manuscript describing an important new tool that will hopefully benefit many neuroscientists.

Reviewer #3 (Remarks to the Author):

The authors appropriately revised and corrected the description in their manuscript according to the reviewer comments, which made their work suitable for publication in Commun. Biol. This reviewer recommends accepting for the publication.